# Uni-RLHF: Universal Platform and Benchmark Suite for Reinforcement Learning with Diverse Human Feedback

**Yifu Yuan**[1], **Jianye Hao**[1,*] **Yi Ma**[1], **Zibin Dong**[1], **Hebin Liang**[1], **Jinyi Liu**[1], **Zhixin Feng**[1]
**Kai Zhao**[1], **Yan Zheng**[1]
{yuanyf,jianye.hao,mayi,zibindong,lianghebin,jyliu,
jingzhe135,kaizhao,yanzheng}@tju.edu.cn
[1]College of Intelligence and Computing, Tianjin University

## ABSTRACT

Reinforcement Learning with Human Feedback (RLHF) has received significant attention for performing tasks without the need for costly manual reward design by aligning human preferences. It is crucial to consider diverse human feedback types and various learning methods in different environments. However, quantifying progress in RLHF with diverse feedback is challenging due to the lack of standardized annotation platforms and widely used unified benchmarks. To bridge this gap, we introduce **Uni-RLHF**, a comprehensive system implementation tailored for RLHF. It aims to provide a complete workflow from *real human feedback*, fostering progress in the development of practical problems. Uni-RLHF contains three packages: 1) a universal multi-feedback annotation platform, 2) large-scale crowdsourced feedback datasets, and 3) modular offline RLHF baseline implementations. Uni-RLHF develops a user-friendly annotation interface tailored to various feedback types, compatible with a wide range of mainstream RL environments. We then establish a systematic pipeline of crowdsourced annotations, resulting in large-scale annotated datasets comprising more than 15 million steps across 30 popular tasks. Through extensive experiments, the results in the collected datasets demonstrate competitive performance compared to those from well-designed manual rewards. We evaluate various design choices and offer insights into their strengths and potential areas of improvement. We wish to build valuable open-source platforms, datasets, and baselines to facilitate the development of more robust and reliable RLHF solutions based on realistic human feedback. The website is available at https://uni-rlhf.github.io/.

## 1 INTRODUCTION

Reinforcement learning (RL) in decision making has demonstrated promising capabilities in many practical scenarios (Feng et al., 2023; Kaufmann et al., 2023; Ni et al., 2023). However, designing dense and comprehensive reward signals for RL requires significant human efforts and extensive domain knowledge (Zhu et al., 2020; Hadfield-Menell et al., 2017; Hu et al., 2020). Moreover, agents can easily exploit designed rewards to achieve high returns in unexpected ways, leading to reward hacking phenomena (Skalse et al., 2022). RLHF (Ibarz et al., 2018; Christiano et al., 2017; Knox & Stone, 2009) serves as a powerful paradigm that transforms human feedback into guidance signals. It eliminates the need for manual reward function design, playing an important role in scenarios such as robotics control (Hiranaka et al., 2023) and language model training (Ouyang et al., 2022). Beyond the reward modeling, the integration of diverse human feedback into reinforcement learning (Ghosal et al., 2023; Guan et al., 2021a) has become the spotlight, as it can align the agent's behavior with the complex and variable intentions of humans (Lindner & El-Assady, 2022; Casper et al., 2023).

Despite the increasing significance of RLHF, the scarcity of universal tools and unified benchmarks specifically designed for RLHF hinders a comprehensive study into learning from various types

---

*Corresponding authors: Jianye Hao (jianye.hao@tju.edu.cn)

of feedback. Several RLHF benchmarks (Lee et al., 2021b; Shin et al., 2023; Daniels-Koch & Freedman, 2022) for comparative feedback have been proposed, but they all utilize predefined hand-engineered reward functions from the environment as scripted teachers, conducting experiments under conditions that simulate absolute expert and infinite synthetic feedback. However, in reality, scenarios akin to unbiased expert feedback are scarce. There is a growing use of crowdsourced annotations (Ouyang et al., 2022; Touvron et al., 2023) as a substitute to cope with the increasing scale of feedback. A significant issue is general unreliability and confusion among large-scale crowdsourced annotators, leading to cognitive inconsistency and even adversarial attacks (Lee et al., 2021b; Bıyık et al., 2019; Ghosal et al., 2023). Therefore, studying the RLHF algorithm design with human teachers instead of script teachers is more compatible with the practical problem. Moreover, the absence of reliable annotation tools that can support diverse types of feedback across environments, as well as efficient pipelines for crowdsourcing, compounds the challenges of conducting research on human feedback.

To bridge the gap, we introduce **Uni-RLHF**, a comprehensive system implementation for RLHF, comprising of three packages: 1) universal annotation platform supporting diverse human feedback for decision making tasks, 2) large-scale and reusable crowdsourced feedback datasets, and 3) modular offline RL baselines from human feedback. Uni-RLHF offers a streamlined interface designed to facilitate the study of learning across a variety of feedback types. Uni-RLHF offers a consistent API for different types of feedback. It uses a query sampler to load and visualize a set of available segments from the datasets. The annotators then provide appropriate feedback labels, which are transformed into a standardized format. Uni-RLHF is compatible with mainstream RL environments and demonstrates strong scalability.

Crowdsourced label collection is often challenging and costly. When gathering human feedback labels instead of script labels, various factors such as different experimental setups, and the number of annotators can introduce significant variability in final performance. Therefore, a standardized data collection process and reliable feedback datasets can greatly alleviate the burden on researchers. We have established a systematic pipeline including crowdsourced annotation and data filters, providing 30 feedback datasets ($\approx$15M steps) of varying difficulty levels that can be reused.

Furthermore, we provide Offline RLHF baselines that integrate mainstream reinforcement learning algorithms and different feedback modeling methods. We then evaluate design choices and compare the performance of baseline algorithms on the collected datasets. We empirically found that Uni-RLHF achieves competitive performance with well-designed manual task reward functions. This offers a solid foundation and benchmark for the RLHF community.

Our contributions are as follows:

- We introduce a universal RLHF annotation platform that supports diverse feedback types and corresponding standard encoding formats.
- We develop a crowdsourced feedback collection pipeline with data filters, which facilitates the creation of reusable, large-scale feedback datasets, serving as a valuable starting point for researchers in the field.
- We conduct offline RL baselines using collected feedback datasets and various design choices. Experimental results show that training with human feedback can achieve competitive performance compared to well-designed reward functions, and effectively align with human preferences.

## 2 RELATED WORK

**Reinforcement Learning from Human Feedback** (RLHF) leverages human feedback to guide the learning process of reinforcement learning agents (Akrour et al., 2011; Pilarski et al., 2011; Wilson et al., 2012; Liu et al., 2024; Zhou et al., 2024), aiming to address the challenges in situations where environment information is limited and human input is required to enhance the efficiency and performance (Wirth et al., 2017; Zhang et al., 2019). RLHF covers a variety of areas such as preference-based RL (Christiano et al., 2017), inverse RL (Abbeel & Ng, 2004; Ng et al., 2000) and active learning (Chen et al., 2020) with various scopes of feedback types. Comparative feedback (Christiano et al., 2017; Lee et al., 2021a; Park et al., 2022; Liang et al., 2022) facilitate a pairwise comparison amongst objectives, culminating in a preference for trajectories, and further transform these rankings into the targets of the reward model. The TAMER (Knox & Stone, 2009; 2012; Warnell et al., 2018; Cederborg et al., 2015) and COACH (MacGlashan et al., 2017; Arumugam et al., 2019) series frameworks furnishes quantifiable evaluative feedback (binary right/wrong signals) on indi-

vidual steps. Despite the excellent generality of comparison and evaluation, the power of expression is lacking (Guan et al., 2023). Recent works (Guan et al., 2021b; Teso & Kersting, 2019) have employed visual explanation feedback encoding task-related features to further broaden the channels of human-machine interaction. Atrey et al. (2019); Liang et al. (2023) enhances learning efficiency through the incorporation of a saliency map provided by humans. Besides, language (Sharma et al., 2022) and relative attribute (Guan et al., 2023) corrective guidance furnishes a richer semantic mode of communication, thereby aligning with user intentions with greater precision.

**Benchmarks and Platforms for RLHF.** Though diverse types of feedback are widely proposed and applied, a prominent limitation is that there are lack of generalized targeted benchmarks for deploying and evaluating different types of human feedback. Freire et al. (2020) proposed a DE-RAIL benchmark for preference-based learning, but it's limited to simple diagnostic tasks. Lee et al. (2021a) considered more complex robotic tasks and designed simulated teachers with irrationality. Shin et al. (2023) evaluated offline preference-based RL on parts of D4RL (Fu et al., 2020) tasks, but these studies primarily focus on synthesizing simulated feedback using scripted teachers. Kim et al. (2023) experimented with small batches of expert feedback and found high disagreement rates between synthesis and human labels, suggesting synthetic labels may be misleading. Besides, Shah et al. (2019) and Ghosal et al. (2023) suggested using real human feedback can improve the learning of human-aligned reward models. In a novel approach, we use large-scale crowdsourced labeling for systematic analysis and experimentation across multiple environments. Another key limitation is the lack of universal platforms for multiple feedback types annotation. While some researchers have open-sourced their tools (Guan et al., 2021a; Kim et al., 2023), they are suitable only for individual, small-scale annotations and specific feedback types, presenting significant challenges for scaling up to a large number of environments. Biyik et al. (2022) provided labeling tools for active learning which are limited in low dimension tasks. A concurrent work, RLHF-Blender (Metz et al., 2023), implemented an interactive application for investigating human feedback, but it lacks scalability for more environments and has not yet been evaluated in a user study.

# 3 Universal Platform for Reinforcement Learning with Diverse Feedback Types

To align RLHF methodologies with practical problems and cater to researchers' needs for systematic studies of various feedback types within a unified context, we introduce the Uni-RLHF system. We start with the universal annotation platform, which supports various types of human feedback along with a standardized encoding format for diverse human feedback. Using this platform, we have established a pipeline for crowdsourced feedback collection and filtering, amassing large-scale crowdsourced labeled datasets and setting a unified research benchmark. The complete system framework is shown in Fig. 1.

## 3.1 Implementation for Multi-feedback Annotation Platform

Uni-RLHF employs a client-server architecture, built upon the Flask[1] back-end service and the Vue[2] front-end framework. It provides an interactive interface for employers and annotators, supporting multi-user annotation. Its intuitive design enables researchers to focus on the task, not the tool. As illustrated in Fig. 1 (upper left), Uni-RLHF packs up abstractions for RLHF annotation workflow, where the essentials include: ❶ interfaces supporting a wide range of online environments and offline datasets, ❷ a query sampler that determines which data to display, ❸ an interactive user interface, enabling annotators to view available trajectory segments and provide feedback responses and ❹ a feedback translator that transforms diverse feedback labels into a standardized format.

**Environment and Datasets Interface.** The Uni-RLHF supports both online and offline training modes. During online training, Uni-RLHF samples data from the buffer periodically and utilizes its built-in render function to present the segments through the user interface. This facilitates asynchronous training of the RL agent and the feedback model. The complete training details are shown in Appendix D. In offline mode, we gather access to several widely recognized reinforcement learning datasets. Some representative tasks from these environments are visualized in Table 1. Furthermore, Uni-RLHF allows easy customization and integration of new offline datasets by simply adding three functions `load_datasets`, `sample` and `visualize`.

---

[1] https://github.com/pallets/flask
[2] https://github.com/vuejs/core

Table 1: Overview of the representative offline datasets supported by Uni-RLHF

| Benchmark | Domain | Agents | Total Tasks |
|---|---|---|---|
| D4RL (Fu et al., 2020) | Gym-Mujoco | Walker2d, Hopper, HalfCheetah | 9 |
| | Antmaze | Antmaze | 6 |
| | Adroit | Door, Pen, Hammer | 9 |
| D4RL_Atari (Seno, 2023) | Atari | - | 26 |
| SMARTS (Zhou et al., 2020) | Automatic Driving | Cruise, Cut-in, Left-turn-cross | 3 |
| V-D4RL (Lu et al., 2022) | DMControl | Walker, Cheetah, Humanoid | 4 |
| MiniGrid (Chevalier-Boisvert et al., 2023) | Four-rooms | - | 2 |

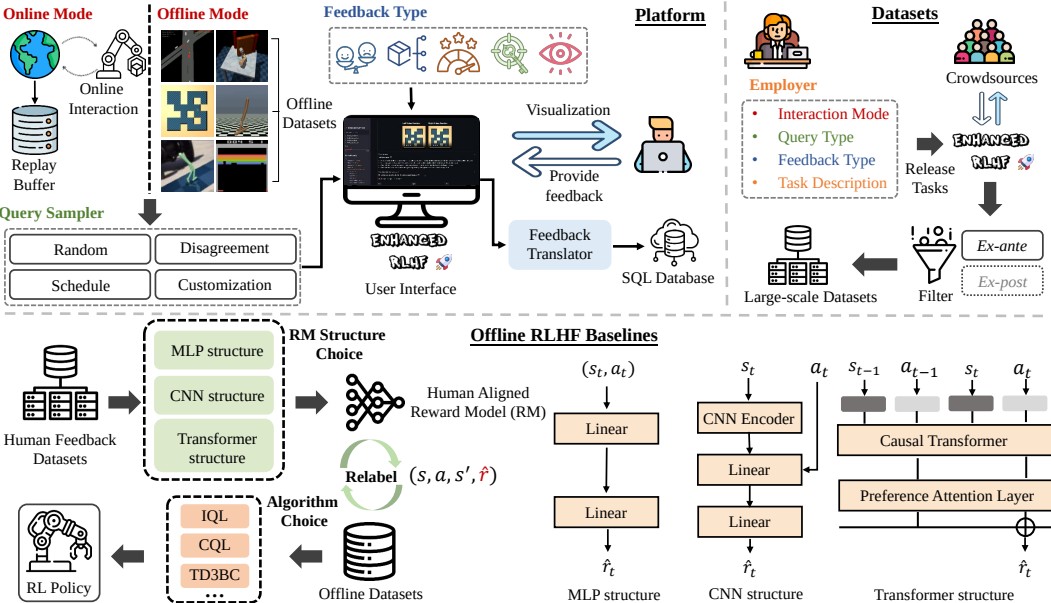

Figure 1: **Overview of the Uni-RLHF system**. Uni-RLHF consists of three components including the platform, the datasets, and the offline RLHF baselines.

**Query Sampler.** The query sampler component determines how to sample the data presented to the user interface. Choosing a suitable query sampler to annotate can help improve model performance (Shin et al., 2023). We propose four configurable types of sampling methods:

- **Random:** Random sampling is the random selection of a specified number. We regard random sampling as the most common baseline, and this simple approach can achieve competitive performance to active learning sampling in numerous environments (Lee et al., 2021b).

- **Disagreement:** Disagreement sampling is primarily applicable for comparative feedback. We calculate the variance of the queries using the probability $p$ that trajectory 1 is superior to trajectory 2 according to the feedback model which encourages the selection of informative queries exhibiting maximum divergence (Christiano et al., 2017).

- **Schedule:** Schedule sampling prefers sampling near on-policy query from the replay buffer during online training, which adapts to the gradually improving policy performance and mitigates the issue of query-policy misalignment (Hu et al., 2023).

- **Customization:** Customized sampling allows for task-specific and customizable metrics sampling that facilitates targeted queries.

**Interactive User Interface.** Uni-RLHF provides users with a clean visual interface, offering an array of interactive modalities for various feedback types. The user interface consists of a segment display, task description, feedback interaction, and additional information module. The feedback interaction supports different levels of granularity, including selection, swiping, dragging, and keyframe capturing. Notably, we also support focused framing of images, which can be applied to the visual feedback, and the annotations can be saved as the PASCAL VOC (Everingham et al., 2010) format. See our homepage for more visualization.

**Feedback Translator.** The feedback translation component receives various feedback from the user interface, which is encoded into consistent feedback labels based on the standardized feedback encoding framework proposed in Section 3.2. Post-processed labels are then matched with queries and linked to a shared storage space, such as an SQL database. This configuration allows the application

to scale, enabling multiple annotators to carry out tasks without interference. This setup facilitates large-scale distributed human experiments.

## 3.2 STANDARDIZED FEEDBACK ENCODING FORMAT FOR REINFORCEMENT LEARNING

To capture and utilize diverse and heterogeneous feedback labels from annotators, we analyze a range of research and propose a *standardized feedback encoding format* along with possible training methodologies. We examine the standard interaction for an RL agent with an environment in discrete time. At each time step $t$, the agent receives state $s_t$ from the environment and chooses an $a_t$ based on the policy $a_t \sim \pi (\cdot \mid s_t)$. Then the environment returns a reward $r_t$ (optional) and the next state $s_{t+1}$. We discuss five common feedback types, explaining how annotators interact with these types and how they can be encoded. Additionally, we briefly outline the potential forms and applications of reinforcement learning that integrate various forms of human feedback. We introduce the encoding format in this section and refer to Appendix F for the corresponding training methods.

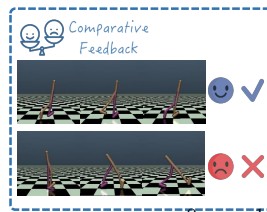

**Comparative Feedback.** Comparative feedback is the most commonly used feedback type for reward modeling because of its simplicity and universality. Crowdsourcing tends to prefer relative comparisons over absolute evaluations. Users are presented with either two trajectory segments for comparison or multiple objectives for ranking. We define a segment $\sigma$ as a sequence of time-indexed observations $\{s_k, ..., s_{k+H-1}\}$ with length $H$. Given a pair of segments $(\sigma^0, \sigma^1)$, annotators give feedback indicating their preference. This preference, denoted as $y_{\text{comp}}$, can be expressed as $\sigma^0 \succ \sigma^1$, $\sigma^1 \succ \sigma^0$ or $\sigma^1 = \sigma^0$. We encode the labels $y_{\text{comp}} \in \{(1,0), (0,1), (0.5, 0.5)\}$ and deposit them as triples $(\sigma^0, \sigma^1, y_{\text{comp}})$ in the feedback datasets $\mathcal{D}$. Following the Bradley-Terry model (Bradley & Terry, 1952), comparative feedback can be utilized to train reward function $\hat{r}_\psi$ that aligns with human preferences. See training details in Appendix F.1.

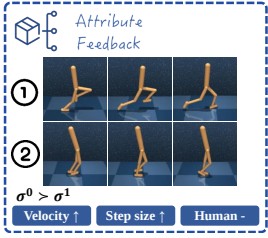

**Attribute Feedback.** In many cases, evaluating the quality of outcomes using a single scalar value is challenging due to the multifaceted nature of numerous problems. For instance, stable robotic movement requires consideration of multiple factors, including speed, stride length, and postural pattern (Zhu et al., 2023). Similarly, autonomous driving necessitates the simultaneous consideration of speed, safety, and rule-adherence (Huang et al., 2021; 2023). Thus an assessment mechanism that evaluates attributes from multiple perspectives is needed to guide the agent towards multi-objective optimization. Inspired by Guan et al. (2023); Dong et al. (2023), we have extended comparative feedback into attribute feedback from multi-perspective evaluation. First, we predefine $\boldsymbol{\alpha} = \{\alpha_1, \cdots, \alpha_k\}$ represents a set of $k$ relative attribute about agent's behaviors. Given a pair of segments $(\sigma^0, \sigma^1)$, The annotator will conduct a relative evaluation of two segments for each given attribute, which is encoded as $y_{\text{attr}} = (y_1, ..., y_k)$ and $y_i \in \{(1,0), (0,1), (0.5, 0.5)\}$. We then establish a learnable attribute strength mapping model that translates trajectory behaviors into latent space vectors. This approach simplifies the problem to a binary comparison and we can distill the reward model similar to comparative feedback. See training details in Appendix F.2.

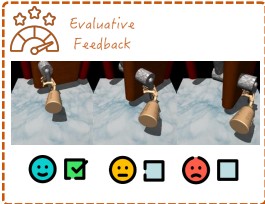

**Evaluative Feedback.** In this feedback mode, annotators observe a segment of a segment $\sigma$ and assign a numerical score or other quantifiable judgment. Evaluative feedback can offer informative absolute evaluations of samples and be less demanding than comparative feedback. Some studies provide feedback to the agent for a specific state, rather than the entire trajectory, potentially leading to a reward-greed dilemma. Therefore, we define $n$ discrete ratings for trajectory levels (White et al., 2023). As shown in the right figure, three ratings are available: good, bad, and average. Thus we receive corresponding human label $(\sigma, y_{\text{eval}})$, where $y_{\text{eval}} \in \{0, ..., n-1\}$ is level of assessment. See training details in Appendix F.3.

**Visual Feedback.** Visual feedback offers a strong way for annotators to interact with agents naturally and efficiently. Annotators identify key task-relevant objects or areas within the image, guiding the agents toward critical elements for decision-making via vi-

sual explanation. We define the feedback as $(\sigma, y_{\text{visual}})$, where $y_{\text{visual}} = \{\text{Box}_1, ..., \text{Box}_m\}$. $\text{Box}_i$ is a the bounding box position $(x_{\min}, y_{\min}, x_{\max}, y_{\max})$. To reduce the burden on the annotators, similar to Gmelin et al. (2023) and Guan et al. (2021b), we provide an object detector that pre-highlights and tracks all objects in the environment, and the annotator only needs to deselect irrelevant objects. We use the detector for atari provided by Delfosse et al. (2023) as an example. As shown in the right figure, all objects including people, a key, monsters, and ladders are pre-framed, so the annotations only need to remove interference targets like ladders. Given an RGB image observation

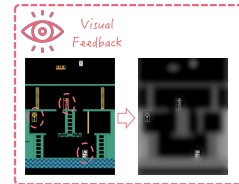

$o_t \in \mathbb{R}^{H \times W \times 3}$, the saliency encoder maps the original input to the saliency map $m_t \in [0, 1]^{H \times W}$ which highlights important parts of the image based on feedback labels. Then the policy is trained with inputs that consist of the RGB images and the saliency predictions as an extra channel. See training details in Appendix F.5.

**Keypoint Feedback.** Frequently, only a few decision points are important in long-horizon tasks (Liu et al., 2023). As illustrated in the right figure, the agent (red arrow) must pick up the key and unlock the door to reach the target location (green circle). Guidance towards key observations such as picking up the key and opening the door can reduce redundant exploration and improve learning efficiency. We annotate key time steps $y_{\text{key}} = \{t_1, t_2, ..., t_n\}$ from segment $\sigma$, where $n$ is the number of keypoints. Subsequently, we use this feedback to train a keypoint pre-

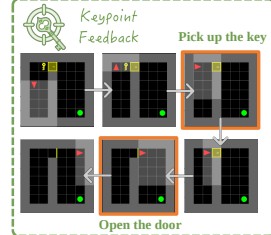

dictor and use historical trajectories to infer key states, which serve as guidance for the agent. See training details in Appendix F.4.

## 3.3 Large-scale Crowdsourced Annotation Pipeline

To validate the ease of use of various aspects of the Uni-RLHF, we implemented large-scale crowd-sourced annotation tasks for feedback label collection, using widely recognized Offline RL datasets. After completing the data collection, we conducted two rounds of data filtering to minimize the amount of noisy crowdsourced data. Our goal is to construct crowdsourced data annotation pipelines around the Uni-RLHF, facilitating the creation of large-scale annotated datasets via parallel crowd-sourced data annotation and filtering. We aim to foster collaboration and reduce the high cost of feedback from RLHF research, facilitating wider adoption of RLHF solutions through reusable crowdsourced feedback datasets. See Appendix I for details of datasets.

**Datasets Collection Workflow.** To generate label and conduct primary evaluations, we assembled a team of about 100 crowdsourced workers, each of whom was paid for their work. These annotators come from a broad array of professions, ranging in age from 18 to 45 and boasting diverse academic backgrounds. None had prior experience in reinforcement learning or robotics research. We aim to simulate a scenario like practical applications of low-cost and non-expert crowdsourced annotation. Using comparative feedback annotation as an example, each annotator is assigned a task along with instructions about the task description and judgment standards. To enhance the understanding of task objectives, experts (authors) provide five example annotations with detailed analysis. We refer to Appendix H for full instructions.

**Ex-ante Filters.** Collecting large-scale feedback often results in suboptimal and noisy data (Luce, 2012). To improve learning performance under human irrationality and bias, we adopt an *ex-ante* approach. During the feedback collection phase, we incorporate a small amount of expert validation set and discard labels that fall below the accuracy threshold. Furthermore, we check labels during the annotation process and summarize the overall problems to optimize the task description. This process aligns the intent of crowd-sourcing with experts. Another approach *ex-post* methods is label noise learning (Wang et al., 2022a; Jeon et al., 2020), which improves performance by understanding and modeling the generation mechanism of noise in the samples. This method is orthogonal to our filtering mechanism.

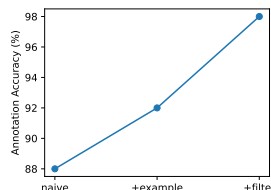

Figure 2: Annotation accuracy in *left-c* task

To verify the effectiveness of each component in the annotation pipeline, we initially sampled 300 trajectory segments of the left-c task in SMARTS for expert annotation, referred to as *oracle*. We

had five crowdsourcing instances, each annotating 100 trajectories in three distinct settings. *naive* implies only seeing the task description, +*example* allows for viewing five expert-provided annotation samples and detailed analysis, and +*filter* adds filters to the previous conditions. The experimental results, displayed in Fig. 2, revealed that each component significantly improved annotation accuracy, ultimately achieving a 98% agreement rate with the expert annotations.

## 4 EVALUATING BENCHMARKS FOR OFFLINE RLHF

Finally, we conducted numerous experiments on downstream decision-making tasks, utilizing the collected crowdsourced feedback datasets to verify the reliability of the Uni-RLHF system. Notably, due to the wide applicability and low crowdsourcing training cost of comparative feedback, we conducted a more detailed study of comparative feedback in Section 4.1, systematically establishing comprehensive research benchmarks. We then performed additional experimental verification of attribute feedback in Section 4.2 and keypoint feedback in Appendix C.5 as a primer. We encourage the community to contribute and explore more extensive baselines and usage for additional feedback types within the Uni-RLHF system.

### 4.1 EVALUATING OFFLINE RL WITH COMPARATIVE FEEDBACK

**Setups.** As shown in Fig. 1 (bottom), we used crowdsourcing comparative feedback for reward learning (see Appendix F.1 for more training details). Then we used the trained reward model to label reward-free offline datasets and train offline RL policy. Because the reward learning process and offline RL were fully decoupled, our methods can simply be combined with any offline RL algorithms. We used three of the most commonly used offline RL algorithms as the backbone, including Implicit Q-Learning (IQL, Kostrikov et al. (2021)), Conservative Q-Learning (CQL, Kumar et al. (2020)) and TD3BC (Fujimoto & Gu, 2021). We adopt implementations of these algorithms from the CORL (Tarasov et al., 2022) library and reuse the architecture and hyperparameters for different environments. To ensure comprehensive evaluation and benchmarking that accurately reflects the capabilities of the offline RLHF algorithm across a wide range of tasks and complexities, we selected a wide range of tasks of varied types, encompassing locomotion, manipulation, and navigation tasks from D4RL (Fu et al., 2020), as well as visual control tasks from D4RL_Atari (Seno, 2023) and realistic autonomous driving scenarios from SMARTS (Zhou et al., 2020). More environment and dataset details can be found in Appendix A. As baselines, we evaluated preference modeling via three reward model architectures, denoted as **MLP, CNN** and **TFM** (Transformer) structure (see Fig. 1 bottom right). MLP and CNN architectures are similar to Christiano et al. (2017); Lee et al. (2021a). MLP is for vector inputs, while CNN is for image inputs. Following the Kim et al. (2023), we also implemented a similar transformer architecture to estimate the weighted sum of non-Markovian rewards. We refer to Appendix B for more details about RM architectures.

#### 4.1.1 D4RL EXPERIMENTS

We measured expert-normalized scores $= 100 \times \frac{\text{score} - \text{random score}}{\text{expert score-random score}}$ underlying original reward from the benchmark. We used **Oracle** to represent models trained using hand-designed task rewards. In addition, we assessed two different methods of acquiring labels: one is crowd-sourced labels obtained by crowd-sourcing through the Uni-RLHF system, denoted as **CS**, and the other is synthetic labels generated by script teachers based on ground truth task rewards, which can be considered as expert labels, denoted as **ST**. The performance is reported in Table 2, *we focus on the performance of algorithms that use crowdsourced labels because they best match the realistic problem setting.*

Based on these results, we can draw several valuable observations. **Firstly,** among the three algorithmic foundations, IQL demonstrated the most consistent and highest average performance across environments when utilizing the same crowd-sourced datasets. This is attributable not only to the high performance of the IQL baseline but also to the consistency of IQL's performance. When comparing the results of IQL-CS and IQL-Oracle, we found that IQL demonstrates competitive performance against Oracle in most of tasks and even surpasses Oracle in some cases. On the contrary, CQL occasionally experiences a policy collapse, leading to a drastic performance drop, as exemplified in the application of the CQL-CS-MLP method to the `hopper-m-r`. **Secondly**, After comparing the results of CS-TFM and CS-MLP (marked as blue), we empirically found that CS-TFM exhibits superior stability and average performance compared to CS-MLP, particularly in the context of sparse reward task settings. We hypothesize that this is due to TFM's modeling approach and its ability to capture key events, which align more closely with human judgment. **Thirdly**, our experiments show that crowd-sourced labels (CS) can achieve competitive performance compared to synthetic labels (ST) in most environments. Especially with CQL and TD3BC backbone, we can

Table 2: Average normalized scores of various RL baselines with human feedback on D4RL datasets. The scores are averaged across 3 seeds. In each dataset, the method which of the CS-MLP and CS-TFM methods is better is highlighted in blue. The best performance of all methods are marked with *. If standard deviation is intersected with the best method, it is also marked with *. For a complete set of results including mean and standard deviation, please refer to Appendix E.

| Dataset | IQL | | | | | CQL | | | | | TD3BC | | | | |
|---|---|---|---|---|---|---|---|---|---|---|---|---|---|---|---|
| | Oracle | ST-MLP | ST-TFM | CS-MLP | CS-TFM | Oracle | ST-MLP | ST-TFM | CS-MLP | CS-TFM | Oracle | ST-MLP | ST-TFM | CS-MLP | CS-TFM |
| walker2d-m | 80.91* | 73.7 | 75.39 | 78.4 | 79.36 | 80.75* | 76.9 | 75.62 | 76.0 | 77.22* | 80.91 | 86.0 | 80.26 | 26.3 | 84.11* |
| walker2d-m-r | 82.15* | 68.6 | 60.33 | 67.3 | 56.52 | 73.09* | -0.3 | 33.18 | 20.6 | 1.82 | 82.15* | 82.8* | 24.3 | 47.2 | 61.94 |
| walker2d-m-e | 111.72* | 109.8 | 109.16 | 109.4 | 109.12 | 109.56* | 108.9 | 108.83 | 92.8 | 98.96 | 111.72* | 110.4 | 110.13 | 74.5 | 110.75* |
| hopper-m | 67.53 | 51.8 | 37.47 | 50.8 | 67.81* | 59.08* | 57.1 | 44.04 | 54.7 | 63.47* | 60.37 | 58.6 | 62.89 | 48.0 | 99.42* |
| hopper-m-r | 97.43* | 70.1 | 64.42 | 87.1 | 22.65 | 95.11* | 2.1 | 2.08 | 1.8 | 52.97 | 64.42* | 44.4 | 24.35 | 25.8 | 41.44 |
| hopper-m-e | 107.42 | 107.7 | 109.16 | 94.3 | 111.43* | 99.26* | 57.5 | 57.27 | 57.4 | 57.05 | 101.17 | 103.7* | 104.14* | 97.4 | 91.18 |
| halfcheetah-m | 48.31* | 47.0 | 45.10 | 43.3 | 43.24 | 47.04* | 43.9 | 43.26 | 43.4 | 43.5 | 48.10 | 50.3* | 48.06 | 34.8 | 46.62 |
| halfcheetah-m-r | 44.46* | 43.0 | 40.63 | 38.0 | 39.49 | 45.04* | 42.8 | 40.73 | 41.9 | 40.97 | 44.84* | 44.2* | 36.87 | 38.9 | 29.58 |
| halfcheetah-m-e | 94.74* | 92.2 | 92.91 | 91.0 | 92.20 | 95.63* | 69.0 | 63.84 | 62.7 | 64.86 | 90.78 | 94.1* | 78.99 | 73.8 | 80.83 |
| mujoco average | 81.63 | 73.7 | 69.9 | 73.29 | 69.09 | 78.28 | 50.9 | 52.09 | 50.14 | 55.65 | 76.45 | 74.8 | 63.33 | 51.86 | 71.76 |
| antmaze-u | 77.00* | 71.59 | 74.67* | 74.22* | 68.44 | 92.75* | 93.71* | 91.71* | 63.95 | 91.34* | 70.75 | 93.51* | 92.90* | 90.25 | 92.30* |
| antmaze-u-d | 54.25 | 51.66 | 59.67 | 54.60 | 63.82* | 37.25* | 34.05 | 25.11 | 6.77 | 22.75 | 44.75 | 73.19* | 36.45 | 51.88 | 59.58 |
| antmaze-m-p | 65.75 | 74.24* | 71.67* | 72.31* | 65.25 | 65.75* | 7.98 | 62.39* | 60.26* | 64.67* | 0.25* | 0.21* | 0.00* | 0.25* | 0.39* |
| antmaze-m-d | 73.75* | 65.74 | 66.00 | 62.69 | 64.91 | 67.25 | 17.50 | 63.27 | 46.95 | 69.74* | 0.25 | 3.33* | 0.39 | 0.10 | 0.32 |
| antmaze-l-p | 42.00 | 40.79 | 43.33* | 49.86* | 44.63* | 20.75 | 1.70 | 18.45 | 44.45* | 19.33 | 0.00 | 0.07* | 0.00 | 0.00 | 0.00 |
| antmaze-l-d | 30.25 | 49.24* | 29.67 | 21.97 | 29.67 | 20.50 | 20.88 | 12.39 | 0.00 | 33.00* | 0.00 | 0.00 | 0.00 | 0.00 | 0.00 |
| antmaze average | 57.17 | 58.91 | 57.67 | 55.94 | 56.12 | 50.71 | 29.3 | 45.55 | 37.06 | 50.14 | 19.33 | 28.38 | 21.62 | 23.75 | 25.43 |
| pen-human | 78.49* | 50.15 | 63.66 | 57.26 | 66.07 | 13.71 | 9.80 | 20.31* | 6.53 | 23.77* | -3.88* | -3.94* | -3.94* | -3.71* | -2.81* |
| pen-cloned | 83.42* | 59.92 | 64.65 | 62.94 | 62.26 | 1.04* | 3.82* | 3.72* | 2.88* | 3.18* | 5.13 | 10.84* | 14.52* | 6.71 | 19.13* |
| pen-expert | 128.05 | 132.85* | 127.29 | 120.15 | 122.42 | -1.41 | 138.34* | 119.60 | 121.14 | 122.41 | 122.53* | 14.41 | 34.62 | 11.45 | 30.28 |
| door-human | 3.26 | 3.46 | 6.8* | 5.05* | 3.22 | 5.53* | 4.68 | 4.92 | 10.31* | 8.81 | -0.13* | -0.32 | -0.32 | -0.33 | -0.32 |
| door-cloned | 3.07* | -0.08 | -0.06 | -0.10 | -0.02 | -0.33* | -0.34* | -0.34* | -0.34* | -0.34* | 0.29* | -0.34 | -0.34 | -0.34 | -0.34 |
| door-expert | 106.65* | 105.35 | 105.05 | 105.72 | 105.00 | -0.32 | 103.90* | 103.32* | 102.63 | 103.15* | -0.33* | -0.34* | -0.33* | -0.34* | -0.34* |
| hammer-human | 1.79* | 1.43* | 1.85* | 1.03* | 0.54* | 0.14 | 0.85* | 1.41* | 0.70* | 1.13* | 1.02* | 1.00* | 0.96* | 0.46 | 1.02* |
| hammer-cloned | 1.50* | 0.70* | 1.87* | 0.67* | 0.73* | 0.30* | 0.28* | 0.29* | 0.28* | 0.29* | 0.25 | 0.25 | 0.27 | 0.45* | 0.35* |
| hammer-expert | 128.68* | 127.4 | 127.36 | 91.22 | 126.50 | 0.26 | 120.16* | 120.66* | 117.65* | 117.60* | 3.11* | 2.22 | 3.13* | 2.13 | 3.03* |
| adroit average | 59.43 | 53.46 | 55.39 | 49.33 | 54.08 | 2.1 | 42.39 | 41.57 | 40.2 | 42.22 | 14.22 | 2.64 | 5.4 | 1.83 | 5.62 |

observe a significant improvement in some cases. This suggests that while synthetic labels are not subject to issues such as noise and irrationality, they are prone to shortsighted behaviors and cannot provide flexible and variable evaluations based on a global view like humans.

### 4.1.2 ATARI EXPERIMENTS

We conducted experiments in five image-based Atari environments, with detailed results presented in Table 3. We collected 2,000 pairs of annotated comparative feedback for each task and trained a reward model for the CNN structure. Next, we chose the CQL of discrete version by d3rlpy[3] codebase and verified the performance of Oracle, ST and CS with the same hyperparameters. Across all tasks, we found that models trained using crowd-sourced labels consistently outperformed

Table 3: Averaged scores of DiscreteCQL on 5 atari tasks with different reward functions. The result shows the average and standard deviation averaged over 3 seeds. We use highlighting to mark the better one of the ST and CS labels.

| Dataset | DiscreteCQL-Oracle | DiscreteCQL-ST | DiscreteCQL-CS |
|---|---|---|---|
| Boxing-m | 78.2±7.8 | 62.4±7.8 | **64.5±10.6** |
| Breakout-m | 79.9±42.0 | 3.0±1.0 | **33.5±10.8** |
| Enduro-m | 196.9±4.9 | 189.5±8.3 | **191.5±8.3** |
| Pong-m | 14.5±2.6 | **18.5±2.2** | 18.6±2.1 |
| Qbert-m | 11013.2±1655.9 | 8170.8±2387.9 | **11301.7±313.3** |

those trained with synthetic labels, achieving performance levels comparable to the Oracle. We hypothesize that this superior performance of crowd-sourced labels could be due to their ability to capture nuanced details within the game environment, details that may prove challenging to encapsulate through simple integral reward functions. This potentially results in more accurate evaluations as compared to those derived from synthetic labels.

### 4.1.3 SMARTS EXPERIMENTS

As shown in Table 4, we studied three typical scenarios in autonomous driving scenarios. The reward function design is particularly complex because it requires more extensive expertise and balancing multiple factors. We

Table 4: Success rate of IQL with different reward model in SMARTS with 3 seeds.

| Task | IQL-Oracle | IQL-ST | IQL-CS |
|---|---|---|---|
| cruise | 0.71±0.03 | 0.55±0.01 | **0.62±0.03** |
| left-c | 0.53±0.03 | 0.52±0.03 | **0.70±0.03** |
| cutin | 0.85±0.04 | **0.81±0.03** | 0.80±0.03 |
| average | 0.70 | 0.63 | **0.71** |

[3]https://github.com/takuseno/d3rlpy

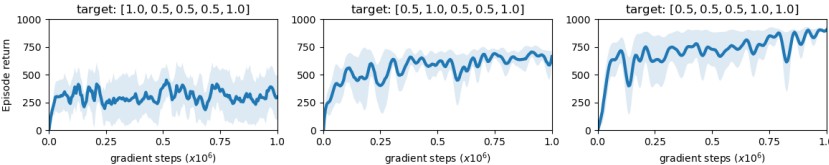

Figure 3: Learning curves of three sub-tasks with different objectives. The target attribute strengths are set to $[1.0, 0.5, 0.5, 0.5, 1.0]$, $[0.5, 1.0, 0.5, 0.5, 1.0]$ and $[0.5, 0.5, 0.5, 1.0, 1.0]$, respectively. The policy can continuously optimize multiple objectives during the training process.

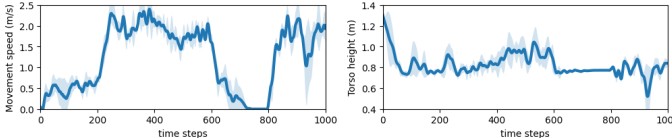

Figure 4: We visualized the behavior switching by adjusting the target attributes every 200 steps. The attribute values for speed were set to $[0.1, 1.0, 0.5, 0.1, 1.0]$, and for height, they were set to $[1.0, 0.6, 1.0, 0.1, 1.0]$. The corresponding changes in attributes can be clearly observed in the curves. We refer to our homepage for the full visualisations of walker.

carefully designed the task rewards incorporating eight factors, which includes some events and multiple penalty (see Table 5 for details). We empirically demonstrate that we can achieve competitive performance simply by crowdsourced annotations, compared to carefully designed reward functions or scripted teachers. We provide a multi-indicator metrics for evaluation (speed and comfort) and visualisation in Appendix C.1. The model using human feedback showed a more conservative and rule-following policy which was consistent with human intentions.

## 4.2 EVALUATING OFFLINE RL WITH ATTRIBUTE FEEDBACK

**Setups.** We trained an attribute-conditioned reward model on attribute feedback datasets for multi-objective optimization (see Appendix F.2 for more training details). To utilize this reward model for training, we adopted an attribute-conditioned TD3BC approach. Specifically, based on the original version of TD3BC, we concatenated the target relative attribute strengths to the inputs of both the policy and value networks. During training, we first evaluated the attribute strengths of the trajectories in the datasets, then obtained the rewards corresponding to each time step on the trajectories using the reward model, and finally used evaluated attribute strengths and rewards to train TD3BC.

**Results.** In our experiments, we trained TD3BC on the walker environment from DMControl, of which pre-defined attributes are (`speed, stride, leg preference, torso height, humanness`) (See Appendix H.2 for more details of attribute definition and feedback collection). We selected three sets of target attribute strengths, representing objectives of making the walker move faster, have a larger stride, or have a higher torso. During training, we validated the algorithm's performance on these three sub-tasks and showed the learning curve in Fig. 3. The results showed that using the attribute-conditioned reward model could indeed achieve multi-objective training. We further tested the trained model's ability to switch behaviors. We ran the model for 1000 steps and adjusted the target attributes every 200 steps, recording the walker's speed and torso height in Fig. 4. The results showed that the policy trained using the attribute-conditioned reward model could adjust its behavior based on the given target objectives, achieving the goal of multi-objective optimization.

## 5 CONCLUSION, CHALLENGE, AND FUTURE DIRECTIONS

To alleviate the difficulties of widely adopting RLHF solutions in applications, we propose a comprehensive system Uni-RLHF. We first implement the platform that supports multiple feedback types along with a feedback standard encoding format. Subsequently, we carry out large-scale crowdsourced annotation creating reusable offline feedback datasets. Finally, we integrate mainstream offline reinforcement learning algorithms with human feedback, evaluate the performance of baseline algorithms on the collected datasets, and establish a reliable benchmark. Although We attempted to cover a wide range of factors in RLHF, there are some limitations. For example, we conduct large-scale experiments with only the most common Comparative Feedback and we lack deep analyses of human irrationality. We summarize some future research directions in Appendix G.

## ACKNOWLEDGMENTS

This work is supported by the National Natural Science Foundation of China (Grant Nos. 92370132), the National Key R&D Program of China (Grant No. 2022ZD0116402) and the Xiaomi Young Talents Program of Xiaomi Foundation. The authors thank Peilong Han for his help in building and deploying the platform.

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

## A  ENVIRONMENT AND DATASETS DETAILS

In this section, we describe the details of all environments and corresponding tasks.

### A.1  THE DETAILS OF THE DATASETS FOR D4RL

**Gym Mujoco** tasks consist of the HalfCheetah, Hopper, and Walker2d datasets from the D4RL (Fu et al., 2020) offline RL benchmark. Walker2d is a bipedal robot control task, where the goal is to maintain the balance of the robot body and move as fast as possible. Hopper is a single-legged robot control task where the goal is to make the robot jump as fast as possible. Halfcheetah is to make a simulated robot perform a running motion that resembles the movement of a cheetah while trying to maximize the distance traveled within a fixed time period. We use the medium, medium-expert, and medium-replay datasets in our evaluations, which consist of (a mixture of different) policies with varying levels of optimality. We use the v2 version of the environment in our experiments.

**AntMaze.** involves controlling an 8-DoF ant quadruped robot to navigate through mazes and reach a desired goal. The agent receives sparse rewards of +1/0 based on whether it successfully reaches the goal or not. We study each method using the following datasets: large-diverse, large-play, medium-diverse, medium-play, umaze-diverse, and umaze. The difference between diverse and play datasets is the optimality of the trajectories they contain. The diverse datasets consist of trajectories directed towards random goals from random starting points, whereas the play datasets comprise trajectories directed towards specific locations that may not necessarily correspond to the goal. We used the v2 version of the environment in our experiments.

**Adroit** contains a series of continuous and sparse-reward robotic environments to control a 24-DoF simulated Shadow Hand robot to twirl a pen, hammer a nail, open a door, or move a ball. This domain was selected to measure the effect of narrow expert data distributions and human demonstrations on a sparse reward, high-dimensional robotic manipulation task. There are three types of datasets for each environment: expert, human, and cloned. We use the v1 version of the environment in our experiments.

### A.2  THE DETAILS OF THE DATASETS FOR ATARI

We evaluate offline RLHF algorithms on five image-based Atari games (**Pong, Breakout, Qbert, Boxing and Enduro**), using the dataset released by D4RL_Atari[4]. The dataset for each task was categorized into three levels of difficulty: mixed denotes datasets collected at the first 1M steps, medium denotes datasets collected between 9M steps and 10M steps, and expert denotes datasets collected at the last 1M steps. We evaluate the algorithms using the medium difficulty dataset because the expert dataset only requires good imitation performance without reward function learning.

### A.3  THE DETAILS OF SMARTS ENVIRONMENTS AND DATASETS COLLECTION

Scalable Multi-Agent Reinforcement Learning Training School (SMARTS)  (Zhou et al., 2020) is a simulation platform designed for deep learning applications in the domain of autonomous driving, striving to provide users with realistic and diverse driving interaction scenarios.

In this study, we selected three representative and challenging driving scenarios from the Driving SMARTS 2022 benchmark (Rasouli et al., 2022). Each scenario corresponds to a distinct driving situation and demands specific driving skills for appropriate handling. Within these scenarios, the infrastructure is hierarchically organized into road, lane, and waypoint structures. At the highest level, the transportation network is structured as a graph where each node represents an individual road. Each road can further encompass one or multiple lanes, and subsequently, each lane is subdivided into a series of waypoints.

Upon environment initialization, given a start point and goal, the simulation environment generates a road-level global driving route for the agent. The timestep for these scenarios is uniformly set at 0.1s. At each timestep, the agent, using the global route, road features, and the dynamic states of the ego vehicle and neighboring vehicles, decides on actions that lead the ego vehicle to accelerate, decelerate, or change lanes. The environment then updates the states of both the ego vehicle and

---

[4]https://github.com/takuseno/d4rl-atari

Table 5: Task Reward Function Design of Smarts

| Item | Value | Definition |
|---|---|---|
| Distance-Traveled Reward | Single-step travelling distance | Encouraging vehicles to travel towards the goal |
| Reach Goal Reward | +20 | If ego vehicle reach the goal |
| Near Goal Reward | $\min(dis\_to\_goal/5, 10)$ | If ego vehicle close the goal |
| Collision Penalty | -40 | Collision |
| Mistake Penalty | -6 | Ego vehicle triggers off road, off lane or wrong way |
| Lane Selection Reward | +0.3 | The ego vehicle is in the right lane |
| Lane Change Penalty | -1 | The ego vehicle changes lane |
| Time Penalty | -0.3 | Each time step gives a fixed penalty term |

other vehicles in the environment based on the agent's decision and other influential factors, such as the behavior of other vehicles, ensuring real-time dynamic information provision for the agent.

The specifics of the three selected scenarios within SMARTS are described as follows:

- **cruise:** The ego vehicle is tasked to travel on a three-lane straight road. During its journey, it must adjust its speed and change lanes when necessary to avoid potential collisions, ensuring a successful arrival at the end of a specified lane.

- **cut-in:** The ego vehicle is tasked to travel on a three-lane straight road where it encounters distinct challenges compared to the standard cruise scenario. Environmental vehicles in this setting may display more aggressive behaviors like lane-changing or sudden deceleration, aiming to obstruct the ego vehicle's progression. As a result, the ego vehicle must adeptly navigate these challenges to ensure a safe arrival at its designated goal.

- **left-c:** The ego vehicle is tasked to commence its journey from a single-lane straight road, progress towards an intersection, and execute a left turn. Following the turn, it is expected to merge into a designated lane on a two-lane straight road.

**Datasets collection.** In each of the three driving scenarios, we employed online reinforcement learning algorithms to train two agents, each designed specifically for the respective scenario. The first agent demonstrates medium driving proficiency, achieving a success rate ranging from 40% to 80% in its designated scenario. In contrast, the second agent exhibits expert-level performance, attaining a success rate of 95% or higher in the same scenario. For dataset construction, 800 driving trajectories were collected using the intermediate agent, while an additional 200 were gathered via the expert agent. By integrating the two datasets, we compiled a mixed dataset encompassing 1,000 driving trajectories.

**Task Reward Function Design.** To direct the ego vehicle efficiently to its goal while prioritizing safety, We went through a number of tweaks and iterations, and eventually established a multifaceted reward function, marked as **Oracle** in Table 4.

This function is consistently applied across the three distinct driving scenarios. The function encompasses several components: distance-traveled reward, reach/near goal reward, mistake event penalty, lane-selection reward, and time penalty. The specifics of each component are detailed below and displayed in the Table 5:

- **Distance-Traveled Reward**: This reward promotes the forward progression of the ego vehicle towards its goal. It is quantified by the distance the vehicle travels along the lane that directs it towards the goal.

- **Reach/Near Goal Reward**: This is a sparse reward, allocated exclusively at the terminal step. Should the ego vehicle successfully reach the goal, a reward of +20 is granted. Furthermore, if the ego vehicle concludes its task within a 20m proximity of the goal, a "near-goal" reward is dispensed, determined as the lesser value between $dis\_to\_goal/5$ and 10. This structure accentuates the importance of closely approaching the goal, fostering an active drive towards the destination.

- **Mistake Event Penalty**: This penalty addresses undesirable actions undertaken by the ego vehicle. Specifically, a significant penalty of -40 is levied if the ego vehicle collides with another vehicle. For individual transgressions, such as coming into contact with road edges or road shoulders, or instances of driving against the flow of traffic, a penalty of -6 is imposed.

- **Lane-Selection Reward**: This encapsulates both a reward and a counteracting penalty. The positive reward, valued at +0.3, is conferred whenever the ego vehicle operates within the lane that

leads it to the goal, thus encouraging correct lane-changing conducive to the goal's direction. Conversely, a penalty of -1 aims to inhibit unwarranted or frequent lane-changing actions, imposed on each instance the ego vehicle transitions between lanes.

- **Time Penalty**: To emphasize efficiency and motivate the vehicle to swiftly reach its destination, a consistent minor penalty of -0.3 is applied at every timestep.

## B    REWARD MODEL STRUCTURE AND LEARNING DETAILS

We implemented the structure of the three reward models with the following implementation details:

**MLP:** We use a three-layer neural network with 256 hidden units each using ReLU as an activation function. To improve the stability in reward learning, we use an ensemble of three reward models and constrain the output using the tanh function. Each model is trained by optimizing the cross-entropy loss defined in Eq. (4) using Adam optimizer with the initial learning rate of 0.0003.

**CNN:** We make the following changes to the MLP structure: 1) We process the image input using a 3-layer CNN with kernel sizes $(8, 4, 3)$, stride $(4, 2, 1)$, and $(32, 64, 64)$ filters per layer. 2) We empirically demonstrate that the tanh activation function exhibits instability in visual environments and is prone to gradient vanishing, so we do not use the activation function for the output. 3) We use a lower learning rate of 0.0001.

**Transformer:** For our experiments, we modified the publicly available codebase of Preference Transformer (PT)[5] and re-implemented our transformer-based reward model. Unlike other structures, PT is better able to capture key events in the long-term behaviors of agents through a preference predictor $P\left[\sigma^1 \succ \sigma^0\right]$ based on the weighted sum of non-Markovian rewards:

$$P\left[\sigma^1 \succ \sigma^0; \psi\right] = \frac{\exp\left(\sum_t w\left(\left\{\left(\mathbf{s}_i^1, \mathbf{a}_i^1\right)\right\}_{i=1}^H; \psi\right)_t \cdot \hat{r}\left(\left\{\left(\mathbf{s}_i^1, \mathbf{a}_i^1\right)\right\}_{i=1}^t; \psi\right)\right)}{\sum_{j\in\{0,1\}} \exp\left(\sum_t w\left(\left\{\left(\mathbf{s}_i^j, \mathbf{a}_i^j\right)\right\}_{i=1}^H; \psi\right)_t \cdot \hat{r}\left(\left\{\left(\mathbf{s}_i^j, \mathbf{a}_i^j\right)\right\}_{i=1}^t; \psi\right)\right)} \quad (1)$$

As shown in Fig. 1 (bottom right), PT consists of **Causal Transformer** and **Preference attention layer** components. In all experiments, we use causal transformers (GPT architecture) (Radford et al., 2018) with one layer and four self-attention heads followed by a bidirectional self-attention layer with a single self-attention head. Given a trajectory state-action segment $\sigma$ of length $H$, the embedding is generated by a linear layer. Then the embedding is fed into the causal transformer network and produces the embedding of the output $\{\mathbf{x}_t\}_{t=1}^H$ where the $t$ th output corresponds to $t$ of the inputs. Then, the preference attention layer receives the hidden embedding from the output of causal transformer $\{\mathbf{x}_t\}_{t=1}^H$ and generates rewards $\hat{r}_t$ and importance weights $w_t$. We define the output $z_i$ as the attention weight values of the $i$ -th query and the keys based on the self-attention mechanism.

$$z_i = \sum_{t=1}^H \text{softmax}\left(\left\{\langle \mathbf{q}_i, \mathbf{k}_{t'}\rangle\right\}_{t'=1}^H\right)_t \cdot \hat{r}_t \quad (2)$$

Then $w_t$ can be defined as $w_t = \frac{1}{H}\sum_{i=1}^H \text{softmax}\left(\left\{\langle \mathbf{q}_i, \mathbf{k}_{t'}\rangle\right\}_{t'=1}^H\right)_t$. Unlike the reward model with MLP structure, PT requires a trajectory as input to generate rewards. Following the codebase of PT, when predicting $r_t$, we take the trajectory of length $H$ up to time $t$ as input and choose the corresponding last output as $r_t$.

Hyperparameters for training transformer structure reward models are shown in Table 6.

## C    ADDITIONAL ANALYSIS AND EVALUATION EXPERIMENTS

### C.1    ADDITIONAL ANALYSIS AND VISUALISATION OF SMARTS EXPERIMENTS

In order to evaluate the performance of the model from multiple perspectives, we have introduced two additional metrics: (1) the average driving speed of the vehicle, to assess the operational efficiency, which is the larger the better, and (2) the comfort level of the vehicle, characterized by the

---

[5]https://github.com/csmile-1006/PreferenceTransformer

Table 6: Hyperparameters for training transformer structure reward models.

| Hyperparameter | Value |
|---|---|
| Embedding dimension | 256 |
| Number of attention heads | 4 |
| Number of layers | 1 |
| Dropout | 0.1 |
| Number of ensembles | 1 |
| Batch size | 64 |
| Optimizer | Adam |
| Learning rate | $10^{-4}$ |
| Weight decay | $10^{-4}$ |

Table 7: **Multi-indicator Evaluation in the SMARTS.** Larger values for success and speed are better, while smaller values for comfort are better.

| | IQL-Oracle | | | IQL-CrowdSource | | |
|---|---|---|---|---|---|---|
| | success rate↑ | speed↑ | comfort↓ | success rate↑ | speed↑ | comfort↓ |
| left-c | 0.53 ± 0.03 | 9.75 ± 0.32 | 7.10 ± 0.06 | **0.70 ± 0.03** | **10.04 ± 0.16** | **6.98 ± 0.26** |
| cruise | **0.71 ± 0.03** | **13.65 ± 0.03** | 1.85 ± 0.08 | 0.62 ± 0.03 | 12.61 ± 0.44 | **1.84 ± 0.49** |
| cutin | **0.85 ± 0.04** | 13.84 ± 0.05 | 0.95 ± 0.23 | 0.80 ± 0.03 | **13.90 ± 0.01** | **0.86 ± 0.05** |
| avg | 0.70 | **12.42** | 3.30 | **0.71** | 12.19 | **3.23** |

sum of the vehicle's linear jerk (the rate of change of linear acceleration) and angular jerk (the rate of change of angular acceleration), which is the smaller the better.

Next, we provide visualization of an instance of the left turn scene in the SMARTS environment. Fig. 5 presents an instance of a left-turn-cross scenario in which the CS model succeeds, while the Oracle model fails due to a collision. In this scenario, the ego vehicle is required to start from a single-lane straight road, make a left turn at the cross intersection, proceed onto another two-lane straight road, and ultimately reach a goal located at the end of one of these two lanes. When navigating the intersection, if there is a stopped or slow-moving neighboring vehicle in front of the ego vehicle, awaiting its turn to pass through, the ego vehicle must decelerate to prevent a collision with the vehicle ahead. The first row of the Fig. 5 shows the driving trajectory of the ego vehicle controlled by the Oracle model. The second row displays the trajectory of the ego vehicle under the control of the CS model. The figure reveals that the Oracle model fails to timely decelerate the ego vehicle, resulting in a collision with the vehicle ahead, whereas the CS model opts to stop and wait for the vehicle in front, eventually passing through the intersection smoothly.

## C.2 ANALYSIS AND VISUALIZATION OF THE LEARNED REWARD MODEL

As shown in Fig. 6, we evaluate the quality of reward model in the training process. Experiments show that the reward model can align with human intent and achieve a high and stable prediction accuracy. The results in Fig. 7 demonstrate that the trained reward model is able to accurately predict step reward for each state and has a strong correlation with the carefully hand-designed reward function.

## C.3 ABLATION OF DIFFERENT QUERY SAMPLER IN OFFLINE RLHF

To better validate the role of different samplers, we collected new crowdsourcing labels and conducted the experiments to demonstrate the performance of different sampling methods. In each experiment, we collect $N_{query} = 100$ annotation labels (comparative feedback) using a different sampler separately and use IQL-MLP-CrowdSourced method for simplicity. Specifically, we consider the following query sampler:

• **Random Sampler:** Randomly select $N_{query}$ pairs of trajectory segments from the dataset.

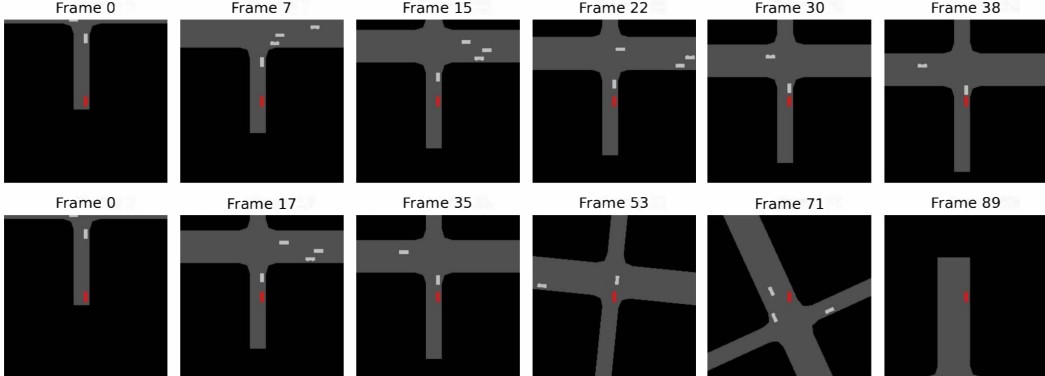

Figure 5: **Visualization an instance of the left turn scene in the SMARTS environment**. The first row shows the driving trajectory of the ego vehicle controlled by the **Oracle** model. The second row displays the trajectory of the ego vehicle under the control of the **CS** model. CS model opts to stop and wait while Oracle model results in a collision.

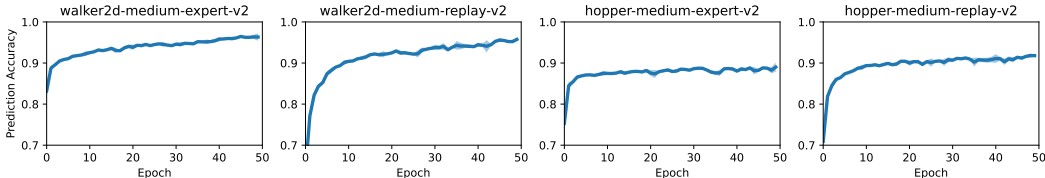

Figure 6: Learning curves of reward model on 4 D4RL MuJoCo tasks. Each experiments was trained with crowdsourced labels and run for 3 seeds.

- **Disagreement Sampler:** Firstly, we generate $N_{init} = 20$ pairs of trajectory segments randomly, then train the initial reward models and measure the variance across ensemble of reward models prediction. Then we select the most $N_{iter} = 40$ high-variance segments. After two iterations, we obtain $N_{query} = 100$ annotation labels and the final reward model is trained.

- **Entropy Sampler:** We modify the customization interface to achieve the entropy sampler, which measure the entropy of preference prediction. The other experiments process is consistent with the disagreement sampler.

Table 8 shows the normalized scores of IQL-MLP-CrowdSourced with various query sampler. We observe that a suitable label sampler could improve the task performance, this is also consistent with a similar conclusion by Shin et al. (2023). The disagreement sampler outperforms random sampling, however, the training phase requires multiple rounds of iterations, which introduces a little additional sampling cost.

## C.4 HUMAN EVALUATION FOR ATTRIBUTE FEEDBACK EXPERIMENTS

RLHF is most useful in domains where only the humans understand the good behaviour like humanness attribute. Therefore, we conducted human evaluation experiments for the agents trained by attribute feedback. We first collected 3 trajectories with different humanness, which were set to [0.1, 0.6, 1] and invited five human evaluators. The human evaluators performed a blind selection to judge which video is most human-like and which video is least human-like. If the result is consistent with the preset labels, it means that the agents can follow the humanness indicator well.

Table 8: Normalized scores of IQL-MLP-CrowdSourced with various query sampler across three seeds.

| Dataset | Random Sampler | Disagreement Sampler | Entropy Sampler |
|---|---|---|---|
| antmaze-medium-diverse-v2 | $46.95 \pm 11.91$ | **64.43±6.57** | **62.69±7.51** |
| hopper-medium-replay-v2 | $42.75 \pm 5.27$ | **63.28±8.58** | $44.53 \pm 6.83$ |

Figure 7: Visualization of the correspondence between predicted reward and ground truth step rewards on 4 D4RL MuJoCo tasks.

Table 9: Normalized scores between TD3BC and TD3BC-keypoint in the maze2d environments. Each run with 3 seeds. We observe that adding keypoint as a guidance achieves significant results in long horizon navigation tasks with sparse reward.

| Dataset | TD3BC | TD3BC-keypoint |
|---|---|---|
| maze2d-medium-v1 | 59.5±36.3 | **131.7±33.3** |
| maze2d-large-v1 | 97.1±25.4 | **150.6±60.2** |

Finally, all people correctly chose the highest humanness trajectory, and only one person chose the lowest humanness trajectory incorrectly. The experimental results confirm that agents through RLHF training are able to follow the abstraction metrics well. We provide video clip in Fig. 10 and full videos on the homepage.

### C.5 EVALUATING OFFLINE RL WITH KEYPOINT FEEDBACK

We trained an keypoint predictor on keypoint feedback datasets in the Maze2d environments (see Appendix F.4 for more training details). Then we adopted an keypoint-conditioned TD3BC approach, where the state of each step treats the keypoint as a subgoal. Compared to the original TD3BC, we concatenated the predicted goal to the inputs of both the policy and value networks. The results are shown in Table 9. TD3BC achieves a significant improvement over baseline because the agents are able to utilize the information provided by the keypoint as a subgoal.

## D  ONLINE RL TRAINING WITH REAL HUMAN FEEDBACK

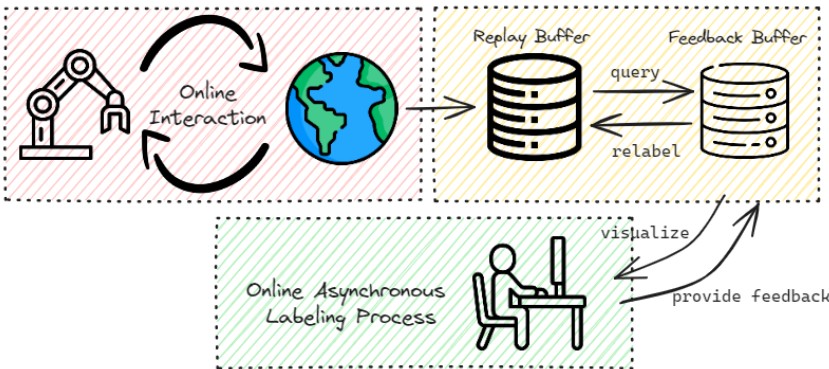

Figure 8: **Online Asynchronous Labeling Process of Uni-RLHF**. We show Uni-RLHF can learn from asynchronous human feedback and annotators interact with agents through remote web interface.

As shown in Fig. 8, Uni-RLHF accommodates the online asynchronous reinforcement learning training process. Much of the previous work (Christiano et al., 2017; Lee et al., 2021a; Park et al., 2022) assumes that online RL training and annotation (even with synthetic labels) are performed alternately, which can lead to low training efficiency for practical applications. Uni-RLHF builds asynchronous pipelines that allow training and annotation to occur simultaneously. It employs a

**Time**

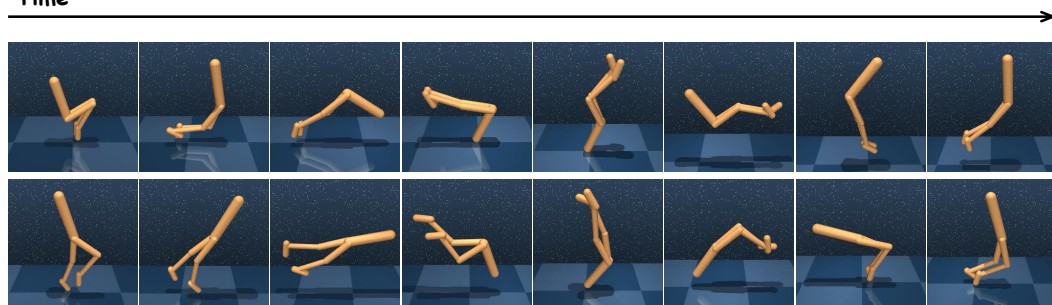

Figure 9: **Multiple front flip demonstration of walker trained via online RLHF**. We show two random case of walker, which is trained using 200 queries of real human feedback. We refer to our homepage for more video visualizations.

sampler to extract data from the replay buffer, which is then stored in the feedback buffer as pending trajectory samples for annotation. Then various annotators can label these samples through the interface without interrupting the training process of the agents. When the number of new annotations reaches a predefined threshold, the reward function undergoes an update, thus relabelling the data stored in the replay buffer.

We validate the effectiveness of the online mode of Uni-RLHF, which allow agents to learn novel behaviors where a suitable reward function is difficult to design. We demonstrate the behaviour of multiple continuous front flip based on the walker environment as shown in Fig. 9. For enabling multiple front flip skill, we train the SAC (Haarnoja et al., 2018) agent with online setting following Lee et al. (2021a). First, we pre-train the exploration policy with an intrinsic reward for the first 2e4 steps to collect diverse behaviors. Collecting a greater diversity of trajectories helps to enhance the value of initial human feedback. Then, at every 1e4 steps, we randomly sample 20 pairs of trajectories from the replay buffer into the feedback buffer to wait for the annotation. The annotators are free to label without interfering with the walker training, and every 50 new labels will retrain the reward model and relabel the reward in the replay buffer. Finally, we give totally 200 queries of human feedback for walker front flips experiments and we observe that walker can master the continuous multiple front flip fluently. The other details and the hyperparameters are the same as Lee et al. (2021a).

# E    FULL RESULTS ON THE D4RL

We report complete average normalized scores on the D4RL datasets. The scores are from the final 10 evaluations (100 evaluations for the antmaze domain) with 3 seeds, representing mean ± std. "-r","-m","-m-r", and "-m-e" is short for random, medium, medium-replay, and medium-expert, respectively. IQL results are presented in Table 10, CQL results are presented in Table 11, and TD3BC results are presented in Table 12.

# F    DETAILS OF STANDARDIZED FEEDBACK ENCODING FORMAT

## F.1    COMPARATIVE FEEDBACK

We define a segment $\sigma$ as a sequence of time-indexed observations $\{\mathbf{s}_k, ..., \mathbf{s}_{k+H-1}\}$ with length $H$. Given a pair of segments $(\sigma^0, \sigma^1)$, a human teacher gives feedback indicating which segment is preferred, so the preference $y_{\text{comp}}$ could be indicated $\sigma^0 \succ \sigma^1$, $\sigma^1 \succ \sigma^0$ or $\sigma^1 = \sigma^0$, which represents equally preferred. We encode the labels $y_{\text{comp}}$ as $y_{\text{comp}} \in \{(1, 0), (0, 1), (0.5, 0.5)\}$ and deposit them as triples $(\sigma^0, \sigma^1, y_{\text{comp}})$ in the feedback $\mathcal{D}$. Subsequently, we treat the preference for trajectories as the training objective to optimize the reward function $\widehat{r}_\psi$. Following the the Bradley-Terry model (Bradley & Terry, 1952), we model human preference based on $\widehat{r}_\psi$ as follows:

$$P_\psi[\sigma^1 \succ \sigma^0] = \frac{\exp(\sum_t \widehat{r}_\psi(\mathbf{s}_t^1, \mathbf{a}_t^1))}{\sum_{i \in \{0,1\}} \exp(\sum_t \widehat{r}_\psi(\mathbf{s}_t^i, \mathbf{a}_t^i))}, \tag{3}$$

Table 10: Average normalized scores on D4RL datasets using IQL.

| Dataset | IQL | | | | |
|---|---|---|---|---|---|
| | Oracle | ST-MLP | ST-TFM | CS-MLP | CS-TFM |
| walker2d-m | 80.91 ± 3.17 | 73.7 ± 11.9 | 75.39 ± 6.96 | 78.4 ± 0.5 | 79.36 ± 2.31 |
| walker2d-m-r | 82.15 ± 3.03 | 68.6 ± 4.3 | 60.33 ± 10.31 | 67.3 ± 4.8 | 56.52 ± 4.62 |
| walker2d-m-e | 111.72 ± 0.86 | 109.8 ± 0.1 | 109.16 ± 0.08 | 109.4 ± 0.1 | 109.12 ± 0.14 |
| hopper-m | 67.53 ± 3.78 | 51.8 ± 2.8 | 37.47 ± 10.57 | 50.8 ± 4.2 | 67.81 ± 4.62 |
| hopper-m-r | 97.43 ± 6.39 | 70.1 ± 14.3 | 64.42 ± 21.52 | 87.1± 6.1 | 22.65 ± 0.8 |
| hopper-m-e | 107.42 ± 7.80 | 107.7 ± 3.8 | 109.16 ± 5.79 | 94.3 ± 7.4 | 111.43 ± 0.41 |
| halfcheetah-m | 48.31 ± 0.22 | 47.0 ± 0.1 | 45.10 ± 0.17 | 43.3 ± 0.2 | 43.24 ± 0.34 |
| halfcheetah-m-r | 44.46 ± 0.22 | 43.0 ± 0.6 | 40.63 ± 2.42 | 38.0 ± 2.3 | 39.49 ± 0.41 |
| halfcheetah-m-e | 94.74 ± 0.52 | 92.2 ± 0.5 | 92.91 ± 0.28 | 91.0 ± 2.3 | 92.20 ± 0.91 |
| mujoco Average | 81.63 | 73.7 | 69.9 | 73.29 | 69.09 |
| antmaze-u-v2 | 77.00 ± 5.52 | 71.59 ± 7.21 | 74.67 ± 5.86 | 74.22 ± 4.22 | 68.44 ± 4.96 |
| antmaze-u-d-v2 | 54.25 ± 5.54 | 51.66 ± 14.04 | 59.67 ± 5.03 | 54.60 ± 4.78 | 63.82 ± 5.60 |
| antmaze-m-p-v2 | 65.75 ± 11.71 | 74.24 ± 4.16 | 71.67 ± 5.77 | 72.31 ± 4.10 | 65.25 ± 4.10 |
| antmaze-m-d-v2 | 73.75 ± 5.45 | 65.74 ± 7.01 | 66.00 ± 2.65 | 62.69 ± 7.51 | 64.91 ± 4.41 |
| antmaze-l-p-v2 | 42.00 ± 4.53 | 40.79 ± 4.89 | 43.33 ± 6.80 | 49.86 ± 8.98 | 44.63 ± 8.20 |
| antmaze-l-d-v2 | 30.25 ± 3.63 | 49.24 ± 6.03 | 29.67 ± 12.66 | 21.97 ± 11.87 | 29.67 ± 9.39 |
| antmaze average | 57.17 | 58.91 | 57.67 | 55.94 | 56.12 |
| pen-human | 78.49 ± 8.21 | 50.15±15.81 | 63.66 ± 20.96 | 57.26±28.86 | 66.07±10.45 |
| pen-cloned | 83.42 ± 8.19 | 59.92 ± 1.12 | 64.65 ± 25.4 | 62.94±23.44 | 62.26±20.86 |
| pen-expert | 128.05 ± 9.21 | 132.85±4.60 | 127.29 ± 11.38 | 120.15±14.78 | 122.42±11.9 |
| door-human | 3.26 ± 1.83 | 3.46 ± 3.24 | 6.8 ± 1.7 | 5.05±4.34 | 3.22±2.45 |
| door-cloned | 3.07 ± 1.75 | -0.08 ± 0.02 | -0.06 ± 0.02 | -0.10±0.018 | -0.016±0.082 |
| door-expert | 106.65 ± 0.25 | 105.35±0.37 | 105.05 ± 0.16 | 105.72±0.20 | 105.00 ± 0.18 |
| hammer-human | 1.79 ± 0.80 | 1.43 ± 1.04 | 1.85 ± 2.04 | 1.03±0.13 | 0.54 ± 0.41 |
| hammer-cloned | 1.50 ± 0.69 | 0.70 ± 0.33 | 1.87 ± 1.5 | 0.67±0.31 | 0.73±0.21 |
| hammer-expert | 128.68 ± 0.33 | 127.4±0.17 | 127.36 ± 0.54 | 91.22±62.33 | 126.5 ± 2.07 |
| adroit average | 59.43 | 53.46 | 55.39 | 49.33 | 54.08 |

Table 11: Average normalized scores on D4RL datasets using CQL

| Dataset | CQL | | | | |
|---|---|---|---|---|---|
| | Oracle | ST-MLP | ST-TFM | CS-MLP | CS-TFM |
| walker2d-m | 80.75 ± 3.28 | 76.9±2.3 | 75.62 ± 2.25 | 76.0 ± 0.9 | 77.22 ± 2.44 |
| walker2d-m-r | 73.09 ± 13.22 | -0.3±0.0 | 33.18 ± 32.35 | 20.6 ± 36.3 | 1.82 ± 0.00 |
| walker2d-m-e | 109.56 ± 0.39 | 108.9±0.4 | 108.83 ± 0.32 | 92.8 ± 22.4 | 98.96 ± 17.1 |
| hopper-m | 59.08 ± 3.77 | 57.1±6.5 | 44.04 ± 6.92 | 54.7 ± 3.4 | 63.47 ± 5.82 |
| hopper-m-r | 95.11 ± 5.27 | 2.1±0.4 | 2.08 ± 0.46 | 1.8 ± 0.0 | 52.97 ± 13.04 |
| hopper-m-e | 99.26 ± 10.91 | 57.5±2.9 | 57.27 ± 2.85 | 57.4 ± 4.9 | 57.05 ± 4.83 |
| halfcheetah-m | 47.04 ± 0.22 | 43.9±0.2 | 43.26 ± 0.25 | 43.4 ± 0.1 | 43.5 ± 0.24 |
| halfcheetah-m-r | 45.04 ± 0.27 | 42.8±0.3 | 40.73 ± 2.14 | 41.9 ± 0.1 | 40.97 ± 1.48 |
| halfcheetah-m-e | 95.63 ± 0.42 | 69.0±7.8 | 63.84±7.89 | 62.7 ± 7.1 | 64.86 ± 8.6 |
| mujoco Average | 78.28 | 50.9 | 52.09 | 50.14 | 55.65 |
| antmaze-u-v2 | 92.75 ± 1.92 | 93.71 ± 1.99 | 91.71 ± 2.89 | 63.95 ± 10.11 | 91.34 ± 3.95 |
| antmaze-u-d-v2 | 37.25 ± 3.70 | 34.05 ± 28.05 | 25.11 ± 10.44 | 6.77 ± 4.09 | 22.75 ± 8.67 |
| antmaze-m-p-v2 | 65.75 ± 11.61 | 7.98 ± 13.07 | 62.39 ± 5.64 | 60.26 ± 9.57 | 64.67 ± 7.32 |
| antmaze-m-d-v2 | 67.25 ± 3.56 | 17.50 ± 6.47 | 63.27 ± 5.61 | 46.95 ± 11.91 | 69.74 ± 2.12 |
| antmaze-l-p-v2 | 20.75 ± 7.26 | 1.70 ± 1.47 | 18.45 ± 6.77 | 44.45 ± 14.00 | 19.33 ± 11.85 |
| antmaze-l-d-v2 | 20.50 ± 13.24 | 20.88 ± 8.55 | 12.39 ± 3.76 | 0.00 ± 0.00 | 33.00 ± 5.57 |
| antmaze average | 50.71 | 29.3 | 45.55 | 37.06 | 50.14 |
| pen-human | 13.71 ± 16.98 | 9.80±14.08 | 20.31±17.11 | 6.53±11.79 | 23.77±14.23 |
| pen-cloned | 1.04 ± 6.62 | 3.82 ± 8.51 | 3.72 ± 7.71 | 2.88±8.42 | 3.18±7.55 |
| pen-expert | -1.41 ± 2.34 | 138.34±5.23 | 119.60±6.45 | 121.14±12.63 | 122.41±10.92 |
| door-human | 5.53 ± 1.31 | 4.68 ± 5.89 | 4.92 ± 6.22 | 10.31±6.46 | 8.81 ± 3.14 |
| door-cloned | -0.33 ± 0.01 | -0.34 ± 0.00 | -0.34 ± 0.00 | -0.34±0.00 | -0.34 ± 0.00 |
| door-expert | -0.32 ± 0.02 | 103.90±0.80 | 103.32±0.22 | 102.63±4.92 | 103.15±5.88 |
| hammer-human | 0.14 ± 0.11 | 0.85 ± 0.27 | 1.41 ± 1.14 | 0.70±0.18 | 1.13 ± 0.34 |
| hammer-cloned | 0.30 ± 0.01 | 0.28 ± 0.01 | 0.29 ± 0.01 | 0.28±0.01 | 0.29±0.01 |
| hammer-expert | 0.26 ± 0.01 | 120.16±6.82 | 120.66±5.67 | 117.65±4.71 | 117.60±2.56 |
| adroit average | 2.1 | 42.39 | 41.57 | 40.2 | 42.22 |

Table 12: Average normalized scores on D4RL datasets using TD3BC

| Dataset | TD3BC | | | | |
|---|---|---|---|---|---|
| | Oracle | ST-MLP | ST-TFM | CS-MLP | CS-TFM |
| walker2d-m | 80.91 ± 3.17 | 86.0±1.5 | 80.26 ± 0.81 | 26.3 ±13.5 | 84.11 ±1.38 |
| walker2d-m-r | 82.15 ± 3.03 | 82.8±15.4 | 24.3 ± 8.22 | 47.2 ± 25.5 | 61.94 ± 11.8 |
| walker2d-m-e | 111.72 ± 0.86 | 110.4±0.9 | 110.13 ± 1.87 | 74.5 ± 59.1 | 110.75 ± 0.9 |
| hopper-m | 60.37 ± 3.49 | 58.6±7.8 | 62.89 ± 20.53 | 48.0 ± 40.5 | 99.42 ± 0.94 |
| hopper-m-r | 64.42 ± 21.52 | 44.4±18.9 | 24.35 ± 7.44 | 25.8 ± 12.3 | 41.44 ± 20.88 |
| hopper-m-e | 101.17 ± 9.07 | 103.7±8.3 | 104.14 ± 1.82 | 97.4 ± 15.3 | 91.18 ± 27.56 |
| halfcheetah-m | 48.10 ± 0.18 | 50.3±0.4 | 48.06 ± 6.27 | 34.8 ± 5.3 | 46.62 ± 6.45 |
| halfcheetah-m-r | 44.84 ± 0.59 | 44.2±0.4 | 36.87 ± 0.48 | 38.9 ± 0.6 | 29.58 ± 5.64 |
| halfcheetah-m-e | 90.78 ± 6.04 | 94.1±1.1 | 78.99 ± 6.82 | 73.8 ± 2.1 | 80.83 ± 3.46 |
| hujoco Average | 76.45 | 74.8 | 63.33 | 51.86 | 71.76 |
| antmaze-u-v2 | 70.75 ± 39.18 | 93.51 ± 2.32 | 92.90 ± 3.08 | 90.25 ± 4.30 | 92.30 ± 1.90 |
| antmaze-u-d-v2 | 44.75 ± 11.61 | 73.19 ± 4.03 | 36.45 ± 5.24 | 51.88 ± 6.52 | 59.58 ± 22.59 |
| antmaze-m-p-v2 | 0.25 ± 0.43 | 0.21 ± 0.33 | 0.00 ± 0.00 | 0.25 ± 0.34 | 0.39 ± 0.57 |
| antmaze-m-d-v2 | 0.25 ± 0.43 | 3.33 ± 1.27 | 0.39 ± 0.55 | 0.10 ± 0.14 | 0.32 ± 0.41 |
| antmaze-l-p-v2 | 0.00 ± 0.00 | 0.07 ± 0.12 | 0.00 ± 0.00 | 0.00 ± 0.00 | 0.00 ± 0.00 |
| antmaze-l-d-v2 | 0.00 ± 0.00 | 0.00 ± 0.00 | 0.00 ± 0.00 | 0.00 ± 0.00 | 0.00 ± 0.00 |
| antmaze average | 19.33 | 28.38 | 21.62 | 23.75 | 25.43 |
| pen-human | -3.88±2.10 | -3.94±0.27 | -3.94±0.24 | -3.71±0.20 | -2.81±2.00 |
| pen-cloned | 5.13±5.28 | 10.84 ± 20.05 | 14.52 ± 16.07 | 6.71±7.66 | 19.13 ± 12.21 |
| pen-expert | 122.53 ± 21.27 | 14.41±14.96 | 34.62±6.79 | 11.45±23.09 | 30.28±14.60 |
| door-human | -0.13±0.07 | -0.32 ± 0.03 | -0.32±0.03 | -0.33±0.01 | -0.32±0.02 |
| door-cloned | 0.29±0.59 | -0.34 ± 0.00 | -0.34±0.00 | -0.34±0.00 | -0.34±0.00 |
| door-expert | -0.33 ± 0.01 | -0.34±0.00 | -0.33 ± 0.01 | -0.34±0.00 | -0.34±0.00 |
| hammer-human | 1.02 ± 0.24 | 1.00 ± 0.12 | 0.96 ± 0.29 | 0.46±0.07 | 1.02±0.41 |
| hammer-cloned | 0.25 ± 0.01 | 0.25 ± 0.01 | 0.27 ± 0.05 | 0.45±0.23 | 0.35±0.17 |
| hammer-expert | 3.11 ± 0.03 | 2.22±1.57 | 3.13 ± 0.02 | 2.13±1.69 | 3.03 ± 0.27 |
| adroit average | 14.22 | 2.64 | 5.4 | 1.83 | 5.62 |

where $\sigma^i \succ \sigma^j$ denotes the event that segment $i$ is preferable to segment $j$ and $(\mathbf{s}_t^i, \mathbf{a}_t^i) \in \sigma^i$. In order to align the reward model with human preferences, the update of $\widehat{r}_\psi$ is transformed into minimizing the following cross-entropy loss:

$$\mathcal{L}(\psi, \mathcal{D}) = -\mathbb{E}_{(\sigma^0, \sigma^1, y_{\text{comp}}) \sim \mathcal{D}} \Big[ y(0) \log P \left[ \sigma^0 \succ \sigma^1 \right] + y(1) \log P \left[ \sigma^1 \succ \sigma^0 \right] \Big], \quad (4)$$

where The term $y(0)$ and $y(1)$ represents the corresponding value in $y_{\text{comp}}$. During the experimental process, the agent can be trained in parallel with the reward model, undergoing iterative updates. It is worth noting that $y_{\text{comp}}$ can simply be extended to a finer-grained division of labels including the degree to which A is better than B (Bai et al., 2022). For example, $y_{\text{comp}} = (0.75, 0.25)$ represents that A is better than B with a smaller degree than (1, 0). Also, the expansion method can simply be trained along the Eq. (4) without any changes.

## F.2 ATTRIBUTE FEEDBACK

First, we predefine $\boldsymbol{\alpha} = \{\alpha_1, \cdots, \alpha_k\}$ represents a set of $k$ relative attributes of the agent's behaviors. Given a pair of segments $(\sigma^0, \sigma^1)$, The annotator will conduct a relative evaluation of two segments for each given attribute, which is encoded as $y_{\text{attr}} = (y_1, ..., y_k)$ and $y_i \in \{(1, 0), (0, 1), (0.5, 0.5)\}$. Then we establish a learnable attribute strength mapping model $\zeta^{\boldsymbol{\alpha}}(\sigma) = \boldsymbol{v}^{\boldsymbol{\alpha}} = [v^{\alpha_1}, \cdots, v^{\alpha_k}] \in [0, 1]^k$ that maps trajectory behaviors to latent space vectors, where $v^{\alpha_i}$ indicates the corresponding attribute. Specifically, the probability that annotators believe that segment $\sigma^0$ exhibits stronger attribute performance on $\alpha_i$ is defined as follows:

$$P_\theta^{\alpha_i}[\sigma_0 \succ \sigma_1] = \frac{\exp \hat{\zeta}_\theta^{\boldsymbol{\alpha}}(\sigma_0)_i}{\sum_{j \in \{1,2\}} \exp \hat{\zeta}_\theta^{\boldsymbol{\alpha}}(\sigma_j)_i}, \quad (5)$$

where $\hat{\zeta}_\theta^{\boldsymbol{\alpha}}(\tau)_i$ represents $i$-th element of $\hat{\zeta}_\theta^{\boldsymbol{\alpha}}$. To facilitate the multi-perspective human intention of the attribute mapping model, we train $\hat{\zeta}_\theta^{\boldsymbol{\alpha}}$ using a modified objective:

$$\mathcal{L}(\hat{\zeta}_\theta^{\boldsymbol{\alpha}}, \mathcal{D}) = - \sum_{i \in \{1, \cdots, k\}} \sum_{(\sigma_0, \sigma_1, y_i) \in \mathcal{D}} y_i(1) \log P^{\alpha_i}[\sigma_0 \succ \sigma_1] + y_i(2) \log P^{\alpha_i}[\sigma_1 \succ \sigma_0] \quad (6)$$

Once such an attribute strength mapping model is established, we can normalize the latent space vectors $\hat{\zeta}_\theta^{\boldsymbol{\alpha}}$ as $(\hat{\zeta}_\theta^{\boldsymbol{\alpha}} - \hat{\zeta}_{\theta,\min}^\alpha)/(\hat{\zeta}_{\theta,\max}^\alpha - \hat{\zeta}_{\theta,\min}^\alpha)$. We then derive evaluations of the trajectories based on the target attributes $\boldsymbol{v}_{\text{opt}}^{\boldsymbol{\alpha}}$ required for the task (relative strength of desired attributes, constrained between 0 and 1), simplifying it into a binary comparison:

$$y(\sigma_0, \sigma_1, \boldsymbol{v}_{\text{opt}}^{\boldsymbol{\alpha}}) = \begin{cases} (1,0), & \text{if } ||\hat{\zeta}_\theta^{\boldsymbol{\alpha}}(\sigma_0) - \boldsymbol{v}_{\text{opt}}^{\boldsymbol{\alpha}}||_2 \leq ||\hat{\zeta}_\theta^{\boldsymbol{\alpha}}(\sigma_1) - \boldsymbol{v}_{\text{opt}}^{\boldsymbol{\alpha}}||_2 \\ (0,1), & \text{if } ||\hat{\zeta}_\theta^{\boldsymbol{\alpha}}(\sigma_0) - \boldsymbol{v}_{\text{opt}}^{\boldsymbol{\alpha}}||_2 > ||\hat{\zeta}_\theta^{\boldsymbol{\alpha}}(\sigma_1) - \boldsymbol{v}_{\text{opt}}^{\boldsymbol{\alpha}}||_2 \end{cases} \tag{7}$$

Subsequently, we establish a conditional reward model $\hat{r}_\psi(\mathbf{s}_t, \mathbf{a}_t, \boldsymbol{v})$ using the pseudo-labels obtained from the transformation. We modified equations Eq. (3) and Eq. (4) to distill the reward model as follows:

$$P_\psi[\sigma^1 \succ \sigma^0 | \boldsymbol{v}^{\boldsymbol{\alpha}}] = \frac{\exp(\sum_t \hat{r}_\psi(\mathbf{s}_t^1, \mathbf{a}_t^1, \boldsymbol{v}^{\boldsymbol{\alpha}}))}{\sum_{i \in \{0,1\}} \exp(\sum_t \hat{r}_\psi(\mathbf{s}_t^i, \mathbf{a}_t^i))}, \tag{8}$$

$$\mathcal{L}(\psi, \mathcal{D}) = -\mathbb{E}_{\boldsymbol{v}^{\boldsymbol{\alpha}} \sim \mathcal{D}} \mathbb{E}_{(\sigma^0, \sigma^1, y_{\text{comp}}) \sim \mathcal{D}} \Big[ y(0) \log P \left[\sigma^0 \succ \sigma^1 \big| \boldsymbol{v}^{\boldsymbol{\alpha}}\right] + y(1) \log P \left[\sigma^1 \succ \sigma^0 \big| \boldsymbol{v}^{\boldsymbol{\alpha}}\right] \Big]. \tag{9}$$

### F.3 Evaluative Feedback

Following the RbRL (Rating-based Reinforcement Learning) (White et al., 2023), we define $n$ discrete ratings for trajectory levels, such as good, bad and average. Therefore we receive corresponding human label $(\sigma, y_{\text{eval}})$, where $y_{\text{eval}} \in \{0, ..., n-1\}$ is level of assessment. Let return $\hat{R}(\sigma) = \sum_t \hat{r}_\psi(\mathbf{s}_t, \mathbf{a}_t)$, we determine rating categories $0 =: \bar{R}_0 \leq \bar{R}_1 \leq \ldots \leq \bar{R}_n := 1$ divided by evaluation distribution. Based on the boundary conditions, we can estimate the probability $P_\sigma(i)$ of each segment being assigned to the $i$-th rating, defined as:

$$P_\sigma(i) = \frac{e^{(\bar{R}(\sigma) - \bar{R}_i)(\bar{R}(\sigma) - \bar{R}_{i+1})}}{\sum_{j=0}^{n-1} e^{(\bar{R}(\sigma) - \bar{R}_j)(\bar{R}(\sigma) - \bar{R}_{j+1})}} \tag{10}$$

Then we use multi-class cross-entropy loss to train the reward model:

$$\mathcal{L}(\psi, \mathcal{D}) = -\mathbb{E}_{(\sigma, y_{\text{eval}}) \sim \mathcal{D}} \Big[ \sum_{i=0}^{n-1} y(i) \log p_\sigma(i) \Big]. \tag{11}$$

### F.4 Keypoint Feedback

We annotate key state time steps $y_{\text{key}} = \{t_1, t_2, ..., t_n\}$ from segment $\sigma$, where $n$ is the number of keypoints. Subsequently, we employ this feedback to train a keypoint predictor $g_\phi$ and use historical trajectories to inference key states, serving as guidance for the agent. Our predictor $g_\phi(\cdot)$ aims to predict the most adjacent keypoints and is trained in supervised manner with a regression loss:

$$\mathcal{L}(\phi, \mathcal{D}) = -\mathbb{E}_{(\sigma, y_{\text{key}}) \sim \mathcal{D}} \Big[ \sum_t \big\| g(\mathbf{s}_t) - \mathbf{s}_{t_{\text{key}}} \big\|_2 \Big], \tag{12}$$

where $t_{\text{key}} \in y_{\text{key}}$ and $\mathbf{s}_{t_{\text{key}}}$ is the keypoint closest to the current $\mathbf{s}_t$. Then the policy are trained with inputs that consist of the states and keypoint predictions as an additional concatenation, marked as $\pi_\theta(s_t, g(\mathbf{s}_t))$. As a result, the agents are able to obtain additional explicit goal guidance during training process.

### F.5 Visual Feedback

We define the feedback as $(s_t, y_{\text{visual}})$, where $s_t \in \sigma$ and $y_{\text{visual}} = \{\text{Box}_1, ..., \text{Box}_m\}$. $\text{Box}_i$ is the bounding box position $(x_{\min}, y_{\min}, x_{\max}, y_{\max})$. Following the Liang et al. (2023), we are able to convert the annotated visual highlights into a saliency map using a saliency predictor $g_\phi$. There have been many saliency prediction model in the literature, e.g. PiCANet (Liu et al., 2018), Deepfix (Kruthiventi et al., 2017). We can then use the human feedback labels to bootstrap the saliency predictor $g_\phi(o_t)$. Given an RGB image observation $o_t \in \mathbb{R}^{H \times W \times 3}$, saliency encoder map the original input to the saliency map $m_t \in [0,1]^{H \times W}$ which highlight important parts of the

image based on feedback labels. There are several methods of aggregating additional saliency visual representations. The CNN encoder and the policy are jointly trained with inputs $(o_t, m_t)$ that consist of the RGB images and saliency predictions as an additional channel. Another possible approach is to use the RGB image $\times$ predicted saliency map as an input. Utilising saliency information enables a more effective focus on relevant and important representations in the image tasks.

## G    SOME CHALLENGES AND POTENTIAL FUTURE DIRECTIONS

We outline some challenges and potential future directions as follows:

**Performance Metrics:** Although we performed evaluation experiments of the RLHF method, it is still evaluated using predefined reward signals, which may not be available in the application. How to quantify human intentions and explain real-world factors is an important research direction.

**Estimations of Human Irrationality and Bias:** Irrationality in human feedback can cause great difficulties in modeling feedback models (Lee et al., 2021b; Wang et al., 2022b; 2017; 2018; 2020), and future research may focus on reducing the negative effects of noise, e.g. noise label learning (Song et al., 2022), and strong model constraints (Xue et al., 2023).

**Imperfect Reward Correction:** Modelling environmental reward through human feedback is easily caused by partially correct rewards (Li et al., 2023). How to second-boost reward modeling through reward shaping (Sorg et al., 2010) and domain knowledge is an area ripe for research.

**Reinforcement Learning with Multi-feedback Integration:** When multiple feedback types are labeled, it is extremely interesting to jointly apply multiple feedback modalities in the same task or to compare different feedback performances.

## H    INSTRUCTIONS FOR CROWDSOURCING ANNOTATORS

In this section, we list the full instructions for each feedback mode for crowdsourcing. In order to improve the consistency of all participant's understanding of the task objectives, for each task, the expert performed 5 annotations as examples and gave a detailed process of analysis. Note that in Atari environments, we also ask the annotators to play for at least ten minutes to better familiarize themselves with the task goals.

### H.1    COMPARATIVE FEEDBACK

The crowdsourced annotators will watch each pair of video clips and **choose which one of the videos is more helpful for achieving the goal of the agent** according to the following task description. If the annotators cannot make a deterministic decision, they are allowed to choose equally.

**Options:**

(1) Choose the left side of the video on the left is more helpful for achieving the goal of the agent.

(2) Choose equal if both videos perform similarly.

(3) Choose the right side of the video on the right is more helpful for achieving the goal of the agent.

---

**Walker2d-medium-v2, Walker2d-medium-replay-v2, Walker2d-medium-expert-v2**

**Description:**
- What you see is two videos of a robot walking. The goal is to move as much to the right as possible while minimizing energy costs.
- If a robot is about to fall or walking abnormally (e.g., walking on one leg, slipping), while the other robot is walking normally, even if the distance traveled is greater than the other robot, it should be considered a worse choice.
- If you think both robots are walking steadily and normally, then choose the better video based on the distance the robot moves.

---

- If both videos show the robots almost falling or walking abnormally, then the robot that maintains a normal standing posture for a longer time or travels a greater distance should be considered the better choice.
- If the above rules still cannot make you decide the preference of the segments, you are allowed to select equal options for both video segments.

## Hopper-medium-v2, Hopper-medium-replay-v2, Hopper-medium-expert-v2

**Description:**
- What you see is a two-part video of a jumping robot. The goal is to move as far to the right as possible while minimizing energy costs.
- If both robots in the two videos are jumping steadily and normally, choose the video based on the distance the robot moves.
- If one robot is jumping normally while the other robot has an unstable center of gravity, unstable landing, or is about to fall, consider the robot with normal jumping as the better choice.
- If both robots in the two videos have experienced situations such as an unstable center of gravity, unstable landing, or about to fall, consider the robot that maintains a normal jumping posture for a longer period or moves a greater distance as the better choice.
- If the above rules still cannot make you decide the preference of the segments, you are allowed to select equal options for both video segments.

## Halfcheetah-medium-v2, Halfcheetah-medium-replay-v2, Halfcheetah-medium-expert-v2

**Description:**
- What you see is two videos of the running of the cheetah robot. The goal is to move as much as possible to the right while minimizing energy costs.
- If both robots are running steadily and normally in the two videos, choose the better video based on the distance the robot moves.
- If one robot is running normally, while the other robot has an unstable center of gravity, flips over, or is about to fall, consider the robot running normally as the better choice.
- If both robots in the two videos have experienced abnormalities, consider the robot that maintains a normal running posture for a longer time or moves a greater distance as the better choice.
- If the above rules still cannot make you decide the preference of the segments, you are allowed to select equal options for both video segments.

## Antmaze-umaze-v2, Antmaze-umaze-diverse-v2, Antmaze-medium-play-v2, Antmaze-medium-diverse-v2, Antmaze-large-play-v2, Antmaze-large-diverse-v2

**Description:**
- What you are seeing are two videos of ants navigating a maze. The objective is to reach the red target position as quickly as possible without falling down.
- If both ant robots are walking normally and making progress toward the target, choose the better video based on the distance to the endpoint.
- If one ant is walking normally while the other ant falls down, stops, or walks in the wrong direction of the target, consider the ant robot walking normally as the better choice.
- If both ants are in abnormal states and there is a significant difference in the distance to the target, consider the ant closer to the target as the better choice.
- If the above rules still cannot make you decide the preference of the segments, you are allowed to select equal options for both video segments.

**Pen-human-v1, Pen-cloned-v1**

**Description:**

- What you see is a dexterous hand whose goal is to manipulate the pen to align its direction and angle with the target pen appearing nearby.

- If in one segment, the angle of the pen in hand eventually becomes consistent with the angle of the target pen nearby, while in another segment, the pen in hand has not yet adjusted to the angle of the target pen, then the former is considered a better choice.

- If both segments have the pen in hand in the process of normal adjustment to the target angle, or both are in abnormal states (stuck, rotating in the wrong direction, etc.), the segment with a direction and angle closer to the target pen is a better choice.

- If the pen in hand is already almost aligned with the direction and angle of the target pen in one segment, while in another segment, the pen in hand is still in the process of adjustment, then the segment that has already achieved the goal is considered a better choice.

- If the above rules still cannot make you decide the preference of the segments, you are allowed to select equal options for both video segments.

**Door-human-v1, Door-cloned-v1**

**Description:**

- What you see is a dexterous hand, the goal is to release the door lock and open the door. When the door contacts the door stop at the other end, the task is considered complete.

- In the two scenarios, if the angle of the door is larger (the maximum angle being the contact with the door stop at the other end), it should be considered better.

- In the two scenarios, if the angle at which the door is opened is the same, then the scenario where the door is opened faster should be considered better.

- If the above rules still cannot make you decide the preference of the segments, you are allowed to select equal options for both video segments.

**Hammer-human-v1, Hammer-cloned-v1**

**Description:**

- What you see is dexterity. The goal is to pick up the hammer and drive the nail into the plank. When the entire length of the nail is inside the plank, the task is successful.

- In the two segments, the deeper the nail is driven into the plank, the better it should be considered.

- In the two segments, if the nail is driven into the plank to the same degree, the scenario that completes the action of nailing the nail more smoothly and seamlessly should be considered better.

- In the two segments, if one segment has nails that are consistently fully hammered in, while the other segment has nails gradually being hammered into the wooden board, then the first one is considered the better choice.

- In the two segments, if a segment experiences anomalies such as the hammer slipping out of hand, it is considered the worse choice.

- If the above rules still cannot make you decide the preference of the segments, you are allowed to select equal options for both video segments.

**Boxing-medium-v0**

**Description:**

- You fight an opponent in a boxing ring. You score points for hitting the opponent. If you score 100 points, your opponent is knocked out.

- You play as a white boxer, the more times you hit the opponent and the fewer times you are hit by the opponent, the better choice it is considered.

- To better understand the game, we recommend that you watch this video[a] and try playing the game for at least 10 minutes using the application[b].

  ___________________________

  [a]https://www.youtube.com/watch?v=nzUiEkasXZI
  [b]https://www.retrogames.cz/zebricky.php

### Breakout-medium-v0

**Description:**

- The dynamics are similar to pong: You move a paddle and hit the ball in a brick wall at the top of the screen. Your goal is to destroy the brick wall. You can try to break through the wall and let the ball wreak havoc on the other side. In two scenarios, scoring more should be considered a better choice.

- To better understand the game, we recommend that you watch this video and try playing the game for at least 10 minutes using the application.

### Enduro-medium-v0

**Description:**

- You are a racer in the National Enduro, a long-distance endurance race. You must overtake a certain amount of cars each day to stay in the race. In two scenarios, scoring more should be considered a better choice.

- To better understand the game, we recommend that you watch this video and try playing the game for at least 10 minutes using the application.

### Pong-medium-v0

**Description:**

- You control the right paddle, you compete against the left paddle controlled by the computer. You each try to keep deflecting the ball away from your goal and into your opponent's goal. The segment of winning more and losing less is considered a better choice.

- To better understand the game, we recommend that you watch this video and try playing the game for at least 10 minutes using the application.

### Qbert-medium-v0

**Description:**

- You are Q*bert. Your goal is to change the color of all the cubes on the pyramid to the pyramid's 'destination' color. To do this, you must hop on each cube on the pyramid one at a time while avoiding nasty creatures that lurk there.

- You score points for changing the color of the cubes to their destination colors or by defeating enemies. In two scenarios, scoring more should be considered a better choice.

- To better understand the game, we recommend that you watch this video and try playing the game for at least 10 minutes using the application.

> **Cruise (Straight road cruising), Cut-in (Dangerous lane change avoidance)**
>
> **Description:**
> What you see is two segments of videos showing autonomous vehicles driving on a straight road with three lanes. The goal is to reach the endpoint located on a specified lane as quickly as possible while ensuring safety. The following rules are prioritized in descending order:
>
> • If one vehicle reaches the endpoint while the other does not, the former is considered better. If one vehicle goes out of bounds or collides while the other does not, the latter is considered better.
>
> • If there is a significant difference in average speed between the two vehicles, i.e., a difference greater than 3m/s, the faster vehicle is considered better.
>
> • If one vehicle consistently stays on the lane where the target point is located while the other does not, the former is considered better.
>
> • If there is no significant difference in both the driving time on the target lane and the average speed of the vehicles (less than 0.2m/s), it is allowed to choose equal options.

> **left-turn-cross (Intersection turning)**
>
> **Description:**
> What you see is two segments of videos showing autonomous vehicles driving in a simulated environment. The goal is to complete the turns as quickly as possible while ensuring safety and reaching the endpoint located on the designated lane. The following rules are prioritized in decreasing order:
>
> • If one vehicle reaches the finish line while the other does not, the former is considered better. If one vehicle goes out of bounds or collides while the other does not, the latter is considered better.
>
> • If the previous rules cannot determine a comparison, the one with a faster average speed is considered better.
>
> • If there is no significant difference in the average speeds of the two vehicles (less than 0.2m/s), it is allowed to choose equal options.

## H.2 ATTRIBUTE FEEDBACK

The crowdsourced annotators will watch each pair of video clips and **determine the relative strength of the performance of the agents** in each attribute based on the behavioral attribute definitions. Select the equal option if the annotators believes that the agents behave consistently on an attribute.

> **DMControl-Walker**
>
> **Options:**
> • Choose the left side if the left video has stronger attribute performance than the right video.
> • Choose equal if both videos perform consistently.
> • Choose the right side if the left video has weaker attribute performance than the right video.
>
> **Definitions of walker attribute:**
> • **Speed**: The speed at which the agent moves to the right. The greater the distance moved to the right, the faster the speed.
> • **Stride**: The maximum distance between the agent's feet. When the agent exhibits abnormal behaviors such as falling, shaking, or unstable standing, weaker performance should be selected.
> • **Leg Preference**: If the agent prefers the left leg and exhibits a walking pattern where the left leg drags the right leg, a stronger performance should be selected. If the agent prefers

> the right leg and exhibits a walking pattern where the right leg drags the left leg, a weaker performance should be selected. If the agent walks with both legs in a normal manner, equal should be selected.
>
> - **Torso height**: The torso height of the agent. If the average height of the agent's torso during movement is higher, stronger performance should be selected. Conversely, weaker performance should be selected if the average height is lower. If it is difficult to discern which is higher, select equal.
> - **Humanness**: The similarity between agent behavior and humans. When the agent is closer to human movement, stronger performance should be selected. When the agent exhibits abnormal behaviors such as falling, shaking, or unstable standing, weaker performance should be selected.

## H.3    KEYPOINT FEEDBACK

**Maze2d-medium-v1, Maze2d-large-v1**

**Instruction:**

Crowdsourced annotators will watch a video clip and accurately label the keypoints in the video based on the task description. keypoints are frames in the video that contain important information or significant scene transitions, such as the appearance of a person, movement of an object, key turning points, or camera switches. Please watch the video carefully and accurately label all the keypoints using the provided tool at the time point where each keypoints appears.

**Description:**

The maze2d domain requires a 2D agent to learn to navigate in the maze to reach a target goal and stay there. You should mark up several keypoints, mainly involving key movements, turns, and other dynamics. If you think the entire video is not helpful for completing the task, you can choose to skip it.

## I    DETAILS OF ANNOTATED FEEDBACK DATASETS

We establish a systematic pipeline of crowdsourced annotations, resulting in large-scale annotated datasets comprising more than 15 million steps across 30 popular tasks. Documentation of the datasets, including detailed information on environments, tasks, number of queries, and length of queries can be found in the following Table 13.

Table 13: Annotated Feedback Datasets details.

| Domain | Env Name | Queries Num | Queries Length | FPS |
|---|---|---|---|---|
| | walker2d-m-v2 | 2000 | 200 | 50 |
| | walker2d-m-r-v2 | 2000 | 200 | 50 |
| | walker2d-m-e-v2 | 2000 | 200 | 50 |
| | hopper-m-v2 | 2000 | 200 | 50 |
| Mujoco | hopper-m-r-v2 | 2000 | 200 | 50 |
| | hopper-m-e-v2 | 2000 | 200 | 50 |
| | halfcheetah-m-v2 | 2000 | 200 | 50 |
| | halfcheetah-m-r-v2 | 2000 | 200 | 50 |
| | halfcheetah-m-e-v2 | 2000 | 200 | 50 |
| | antmaze-umaze-v2 | 2000 | 200 | 50 |
| | antmaze-umaze-diverse-v2 | 2000 | 200 | 50 |
| Antmaze | antmaze-medium-play-v2 | 2000 | 200 | 50 |
| | antmaze-medium-diverse-v2 | 2000 | 200 | 50 |
| | antmaze-large-play-v2 | 2000 | 200 | 50 |
| | antmaze-large-diverse-v2 | 2000 | 200 | 50 |
| | pen-human-v1 | 2000 | 50 | 15 |
| | pen-cloned-v1 | 2000 | 50 | 15 |
| Adroit | door-human-v1 | 2000 | 50 | 15 |
| | door-cloned-v1 | 2000 | 50 | 15 |
| | hammer-human-v1 | 2000 | 50 | 15 |
| | hammer-cloned-v1 | 2000 | 50 | 15 |
| | cutin | 2000 | 50 | 15 |
| Smarts | left_c | 2000 | 50 | 15 |
| | cruise | 2000 | 50 | 15 |
| | breakout-m | 2000 | 40 | 20 |
| | boxing-m | 2000 | 40 | 20 |
| Atari | enduro-m | 2000 | 40 | 20 |
| | pong-m | 2000 | 40 | 20 |
| | qbert-m | 2000 | 40 | 20 |
| Deepmind Control | walker (attribute feedback) | 4000 | 200 | 50 |

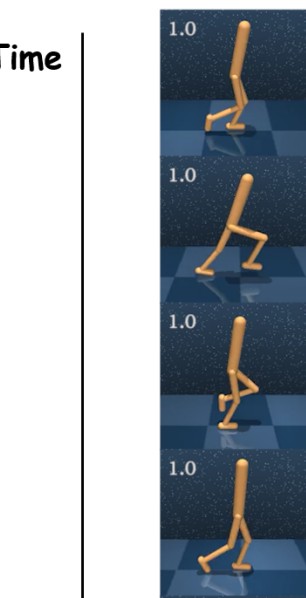

Figure 10: Video clips of different humanness attribute levels. From left to right the target attributes are set to $[1.0, 0.6, 0.1]$.

