# OpenReview forum: "Uni-RLHF: Universal Platform and Benchmark Suite for Reinforcement Learning with Diverse Human Feedback"
_ICLR.cc/2024/Conference — ICLR 2024 poster_

### Official Review · Reviewer_BKHb · 2023-10-13

**Soundness:** 2 fair
**Presentation:** 3 good
**Contribution:** 2 fair
**Rating:** 5
**Confidence:** 2

**Summary:**

This work introduces Uni-RLHF, an eco-system for Reinforcement Learning with Human Feedback to facilitate the data collection with human annotators, the sharing of datasets and the RL alignment.
In particular, the annotation platform named Enhanced-RLHF supports various feedback types.
Then, the paper investigates different offline RL strategies, and show that the reward models trained on their crowdsourcing labels lead to better performances than when using synthetic labels, and can approximate the Oracle rewards. This is done on motion control/manipulation tasks such as D4RL or Atari or Smarts. They aim for fair evaluation of RLHF strategies.

**Strengths:**

- The work effectively succesfully presents a comprehensive system to deal with the data collection process with diverse human feedback types.
- The motivation is clear and interesting: indeed, RLHF appears nowadays as a go-to strategy to ensure the reliability of AI systems. Therefore the proposed eco-system can be of interest to some researchers.
- Crowdsourced labels are sufficient to approximate Oracle-based reward, showing the flexibility of RLHF even in those D4RL datasets.

**Weaknesses:**

- The main weakness is the limitation to control/locomotion tasks. More real world tasks/other modalities (such as text), and also larger architectures are required to make this eco-system more attractive.
- The benchmark only compares offline RL, thus for example the de facto online strategy for RLHF (PPO) is not ablated.
- The different query samplers are not ablated.
- Only the comparative feedback is ablated. It would have been interested to compare the quality of the reward models.
- While the authors do acknowledge this, providing a data cleaning procedure (or data filters) in the eco-system would be very positive to foster its applicability.

**Questions:**

- could you please clarify the differences with RLHF-Blender ? in which case we should use one or the other ?
- How did you select the hyperparameters?
- Have you applied any strategy to refine the dataset quality?

---

> ### Author Response · Authors · 2023-11-22
> **Response to Reviewer BKHb (Part 1/3)**
>
> We thank the reviewer for the insightful and useful feedback, please see the following for our response.
>
> **[Q1: The main weakness is the limitation to control/locomotion tasks. ]**
>
> We would like to clarify that Uni-RLHF primarily focus on RLHF research for **decision making (locomotion/manipulation/navigation…)**, building upon the settings established by a series of mainstream research works such as Preference-PPO[1], Pebble[2]. These works concentrate on decision-making tasks too, and we have developed a labeling platform and benchmark that can accommodate a broader range of decision making tasks.
>
> Same as many widely used annotation tools, e.g. PixelAnnotationTool[3] for vision domain and Curve[4] for time series domain, Uni-RLHF also focuses on specific areas and hopes to build a robust eco-system for decision-making.  Therefore, we are able to focus on solving specific problems in decision tasks annotation, such as a robust environment access methods, feedback format encoding, and so on. Also, we modify relevant description of introduction to avoid confusion and add the statement that we specialise in decision making domain. (**See red text in the updated manuscript, page 1 and 7**)
>
> Based on the above core design concepts, we will also continue to maintain and update the platform to support more real-world problem environments/datasets and more modalities.
>
> **[Q2: The benchmark only compares offline RL, online RLHF is not ablated. ]**
>
> The reason we mainly use Offline RL is that we want to provide researchers with reusable labelled datasets to easily evaluate algorithms without additional labelling cost on a uniform benchmark. While decision making tasks in Online RL needs to learn from scratch, the experimental samples and annotations will be completely different from the algorithms, sampling methods, etc., which makes it difficult to reuse.
>
> As **section 3.1** mentioned, Uni-RLHF is able to support the online training mode. We add a more detailed implementation of the online asynchronous annotation process in the revision. (**see Appendix E and Fig.8, page 21**)
>
> To better show the effectiveness of the online mode of Uni-RLHF, we add an extra online RLHF experiment in the revision (**See Appendix E and Fig.9, page 23**). **After totally 200 queries of human annotation, walker can master the continuous multiple front flip fluently**. Through our experiments we demonstrated that Uni-RLHF can access online training to allow agent to learn novel behaviours that are difficult to design reward.
>
> **[Q3: The different query samplers are not ablated. ]**
>
> Offline RLHF experimental results with crowdsourced annotation are based on the random sampler implementation because of limited resource. However, we provide several ways of sampler definition in our platform to facilitate flexible selection by users and researchers, and **our focus is to provide an open and easily extensible platform to customise the appropriate query according to specific tasks.** Therefore we hope to open up better tools and remain some open research problems to RLHF community like how to choose the sampler.
>
> To better validate the role of different sampler, we collected new crowdsourcing labels and conducted the additional experiments to demonstrate the performance of different sampling methods. In each experiment, we collect **100 annotation labels** (comparative feedback) using a different sampler separately and use IQL-MLP-CrowdSource method for simplicity. **See Appendix D.3 (page 19)** of our revision for detailed experimental setup and results.
>
> | Env Name | Random Sampler | Disagreement Sampler | Entropy Sampler |
> | --- | --- | --- | --- |
> | antmaze-medium-diverse-v2 | $46.95 ± 11.91$ | **64.43 ± 6.57** | **62.69 ± 7.51** |
> | hopper-medium-replay-v2 | $42.75±5.27$ | **63.28 ± 8.58** | 44.53 ± 6.83 |
>
>
> We observe that suitable label sampler could improve the task performance,  this is also consistent with a similar conclusion by OPRL[5]. More ablation experiments are in process and we will continue to update the results in rebuttal phases.

---

> > ### Author Response · Authors · 2023-11-22
> > **Response to Reviewer BKHb (Part 2/3)**
> >
> > **[Q4: Only the comparative feedback is ablated. It would have been interested to compare the quality of the reward models. ]**
> >
> > Thank you for your valuable suggestions! In the current version we think that different feedback types are suitable for different task settings, e.g., comparative feedback is suitable for reward modelling, while attribute feedback for more complex multi-objective tasks and visual feedback for extracting visual representations, and it is very interesting to use a combination or to compare different annotation types in the same task, which we incorporate in our future work. (**See Appendix H, Some challenges and potential future direction**)
> >
> > In the meantime, We have added further analysis and visualisation of the reward model in the revision (**See Appendix D.2, page 19**). We experimentally demonstrated that the reward model trained by human feedback can be well fitted to trends of the hand-designed reward function (**high relevance**) and aligned with human intent (**high prediction accuracy**).
> >
> > **[Q5: Have you applied any strategy to refine the dataset quality?]**
> >
> > Yes, we want to open source high-quality research datasets and have taken the following approaches to improve annotation quality:
> >
> > - **Optimisation of Task Instructions:** We refer to **Appendix J** for full instructions, all instructions were read by annotators unfamiliar with RL before being officially labeled and then optimized. And in gaming environments such as atari, we also let the annotators try it out.
> > - **Annotation Example:** For each task, we provide five well-annotated samples for the annotator to learn, including the how to annotate, annotation results and detailed explanations.
> > - **Validation Set:** We incorporate a small amount of expert validation set and discard labels that fall below the accuracy threshold.
> > - **Sampling Inspection:** We sample the existing annotation for inspection during the annotation process, reject the wrong samples, summarise the overall problem and optimise the task description.
> >
> > **[Q6: providing a data cleaning procedure (or data filters) in the eco-system would be very positive to foster its applicability. ]**
> >
> > That's certainly good advice! As we showed in **section 4.2 Ex-ante Filters and response to the Q5**, adding ex-ante filters can have a partial improvement in annotation quality. Therefore, we have added a **sampling inspection mode** to the eco-system in order to check the annotator's annotation quality in real time.
> >
> > On the other hand, we see the potential for large multi-modal language model to assist human annotation and hope to assist human feedback through RLAIF, which is left for future work.
> >
> > **[Q7: could you please clarify the differences with RLHF-Blender?]**
> >
> > - **More stable functionality:** RLHF-Blender is a project in process. As it is noted at github[6]: *Right now, RLHF-Blender is still in preview mode and might therefore contain bugs or will not immediately run on each system.* The current version of RLHF-blender is more like only providing a UI system without any user study. But Uni-RLHF can support deployments for multi-user annotation and provide rich user study.
> > - **More comprehensive system:** RLHF-Blender only has the core functionality for annotation (in-process), whereas Uni-RLHF provides a range of core functionality practicality associated with the full annotation system, such as task publishing, task binding, viewing annotations and performing sampling inspection.
> > - **Better Scalability:** RLHF-Blender currently supports only a few environments such as pong and minigrid. But Uni-RLHF support dozens of environments across multiple domains and opens up clean interfaces for environments and feedback types.
> > - **Complete eco-systems: Uni-RLHF provide complete eco-systems including** universal annotation platform ,standardized feedback encoding format, large-scale crowdsourced datasets, and modular offline RLHF baselines.
> >
> > Therefore, we recommend the use of Uni-RLHF for human feedback annotation in decision-making tasks.

---

> > > ### Author Response · Authors · 2023-11-22
> > > **Response to Reviewer BKHb (Part 3/3)**
> > >
> > > **[Q8: How did you select the hyperparameters?]**
> > >
> > > We did not carefully select or adjust the hyperparameters for feedback model training, and we only used one set of hyperparameters for each structure (mlp, cnn, transformer).  It also shows that our training method is clean and robust. Full hyperparameters and experimental setups are shown in Appendix B. ****As for RL policy training, we keep the same hyperparameters as original Clean Offline RL[7] implementation for all experiments and do not further fine-tune them.  It's possible that tuning hyperparameters on the human-annoted datasets could bring benefits, but we believe it's orthogonal to our main contribution, which could be one of our future works.
> > >
> > > ---
> > >
> > > [1] Christiano P F, et al. Deep reinforcement learning from human preferences. NIPS2017.
> > >
> > > [2] Lee K, et al. Pebble: Feedback-efficient interactive reinforcement learning via relabeling experience and unsupervised pre-training. ICML, 2021.
> > >
> > > [3] PixelAnnotationTool [https://github.com/abreheret/PixelAnnotationTool](https://github.com/abreheret/PixelAnnotationTool)
> > >
> > > [4] Curve [https://github.com/baidu/Curve](https://github.com/baidu/Curve)
> > >
> > > [5] Shin D, Dragan A, Brown D S. Benchmarks and Algorithms for Offline Preference-Based Reward Learning. Transactions on Machine Learning Research, 2022.
> > >
> > > [6] RLHF-Blender. [https://github.com/ymetz/rlhfblender](https://github.com/ymetz/rlhfblender)
> > >
> > > [7] CORL. [https://github.com/tinkoff-ai/CORL](https://github.com/tinkoff-ai/CORL)

---

### Official Review · Reviewer_FdzJ · 2023-10-30

**Soundness:** 3 good
**Presentation:** 3 good
**Contribution:** 3 good
**Rating:** 6
**Confidence:** 3

**Summary:**

Uni-RLHF is a comprehensive system for reinforcement learning with diverse human feedback. It includes an Enhanced-RLHF platform that supports multiple feedback types, environments, and parallel user annotations, along with a feedback standard encoding format. The system also provides large-scale datasets, offline RLHF baselines, and human feedback datasets for relabeling and human-aligned reward models.

**Strengths:**

1. Uni-RLHF provides a universal platform for RLHF that supports multiple feedback types, environments, and parallel user annotations. The system includes a feedback standard encoding format that facilitates the integration of diverse feedback types.
2. Uni-RLHF provides large-scale datasets and offline RLHF baselines for evaluating the performance of RL algorithms with human feedback. The system also includes human feedback datasets for relabeling and human-aligned reward models, which can improve the efficiency and effectiveness of RLHF.
3. Uni-RLHF can foster progress in the development of practical problems in RLHF by providing a complete workflow from real human feedback.

**Weaknesses:**

1. Uni-RLHF's large-scale crowdsourced feedback datasets may contain noise and bias, which can affect the performance of RL algorithms trained on these datasets.
2.  The system's offline RLHF baselines may not be optimized for specific applications, which can limit their usefulness in practical settings.
3. The system's reliance on human feedback may introduce additional costs and delays in the RL development process, compared to purely synthetic feedback.
Uni-RLHF is an engineering-focused system that emphasizes incremental improvements rather than groundbreaking innovations.

**Questions:**

Please refer to weaknesses

---

> ### Author Response · Authors · 2023-11-22
> **Response to Reviewer FdzJ**
>
> We thank the reviewer for the insightful and useful feedback, please see the following for our response.
>
> **[Q1: Uni-RLHF's large-scale crowdsourced feedback datasets may contain noise and bias, which can affect the performance of RL algorithms trained on these datasets.** **]**
>
> We would like to clarify that noise and bias can't be avoided in pratical problems and annotations,  that's what we're focusing on. We hope to estimate human irrationality and bias in realistic annotators to futher study the RL algorithms in more realistic scenario.
>
> Several RLHF benchmarks[1][2] utilize predefined hand-engineered reward functions from the environment as scripted teachers, which can be considered a noise-free environment. Experimental results in **Table.2** show that **our crowdsourced human feedback can achieve competitive performance or outperform compared to noise-free scripted teachers**.
>
> In the data collection process, we try to minimise extreme noise and refine the dataset quality including continuous optimisation of task instructions, annotation example and sampling inspection.
>
> We have added further analysis and visualisation of the trained reward model (**See Appendix D.2, page 19, revised version paper**) and experimentally demonstrated that the reward model trained by human feedback can be well fitted to trends of the hand-designed reward function (**high relevance**) and aligned with human intent (**high prediction accuracy**).
>
> **[Q2: The system's offline RLHF baselines may not be optimized for specific applications, which can limit their usefulness in practical settings.** **]**
>
> We agree with the reviewer that the addition of more specific applications can increase the usefulness of the Uni-RLHF, which is also in line with our core motivation. Due to the high cost of labelling, previous research has mostly been in more ideal scenarios such as synthetic labels,  however, we provide annotation tools for multi-user annotation and a large number of  crowdsourced labels and this will facilitate the RLHF community to research in more realistic settings. Our main goal is to demonstrate the feasibility of a complete annotation pipeline and an open source reusable dataset. As for offline RLHF baseline, we only utilize the collected crowdsourced feedback datasets to verify the reliability of the Uni-RLHF system, which is not the core contribution of this work. We hope that more researchers will access their real-world problems in the platform to annotate and further optimise the framework based on this codebase.
>
> **[Q3: The system's reliance on human feedback may introduce additional costs and delays in the RL development process, compared to purely synthetic feedback.** **]**
>
> It is worth noting that in the most of RLHF application scenarios, **we do not have access to synthetic feedback. Purely synthetic labels exist only in ideal research simulation environments**, as a wide range of researchers would struggle to afford the cost of crowdsourcing labels in all experiments. The process of generating synthetic labels is as follows: 1) utilize predefined hand-engineered reward functions as scripted teachers, 2) score of trajectories by scripted teachers to obtain synthetic labels. In application scenarios, RLHF is mostly suitable for environments where manual design rewards are difficult, when additional costs and delays are unavoidable.
>
> Take SMARTS domain as an example, designing reward functions is extremely difficult, requiring extensive expertise and multiple rounds of training attempts for iteration, the final training reward (marked as oracle) consists of five reward items of widely varying magnitude: Distance, Near Goal, Mistake Event, Lane-Selection and Time Penalty. Adjusting the reward function also requires a significant amount of cost, whereas crowdsourcing feedback annotations enable simply to achieve competitive performance to oracle and more consistent with human intentions. **(See Section 4.1.3, page 8 and Appendix D.1, page 18 for details)**
>
> ---
>
> [1] Lee K, et al. B-pref: Benchmarking preference-based reinforcement learning. arXiv preprint arXiv:2111.03026, 2021.
>
> [2] Shin D, Dragan A, Brown D S. Benchmarks and Algorithms for Offline Preference-Based Reward Learning. Transactions on Machine Learning Research, 2022.

---

### Official Review · Reviewer_3bfZ · 2023-11-06

**Soundness:** 3 good
**Presentation:** 3 good
**Contribution:** 3 good
**Rating:** 8
**Confidence:** 3

**Summary:**

The paper introduces a new framework to collect human feedback for RLHF. The framework allows for five different types of feedback to be collected with a convenient user interface, depending on the task at hand. A dataset of feedback of three of these types is collected using crowdsourced human workers. This dataset is then used to evaluate multiple existing offline RLHF approaches, where a trained reward model is used to label trajectories from a dataset, and then an offline RL algorithm is used to produce a policy. The evaluation shows that the collected data for comparative and attribute feedback is of good quality, and allows for the learned policy to perform at a quality comparable to a policy trained from hand-crafted reward models. In some cases the policies trained from human feedback outperform those trained from hand-crafted reward models.

**Strengths:**

In my view, the main strength of the paper is the open-sourced dataset and the tool for feedback collection. These will bring great value to the community and allow for easy benchmarking of offline RLHF methods. The extensive evaluation on D4RL is another strong point of the paper. It will provide a reasonable baseline for future offline RLHF approaches. Finally, the dataset with collected attribute feedback can be used for multi-objective learning, an area where fewer datasets in general are available.

**Weaknesses:**

- The claim for a "standardized feedback encoding format along with corresponding training methodologies" in Section 3.2 might be overly ambitious. Most importantly, the training methodologies are only provided in the appendix for two out of the five feedback types. Comparative feedback interface also does not allow the user to indicate the strength of preference, as done e.g. in the Anthropic's dataset for language model alignment [1]. The extension of the methodology in Appendix E.1 to this type seems straightforward (introduce y=(0.75,0.25), for example), and it could be beneficial to mention this.
- Related to the previous point, the paper provides too few details on the way Atari experiments were performed. Appendix G and Section 4.1.1 imply that comparative feedback was collected, but Figure 5 (d) in the appendix -- that visual feedback was used instead. The highlighting in Table 8 should be explained. My guess is that the best of the ST and CS labels is highlighted.
- There is no dataset or benchmarking for evaluative feedback, hence it is hard to assess the usefulness of this part of the system. It is unclear whether data for keypoint and visual feedback on the Atari environments was collected. I believe that the datasets for comparative and attributive feedback are already a good enough contribution, and an interface for other feedback types is a nice-to-have extra, so this point does not make me give this paper a negative rating.
- Star annotations in Table 2 are incomplete. As I understand it, higher is better in the entire table. If that is the case, stars that should mark that the method performs better than the oracle are missing in several places: (hopper-m, CQL, CS-TFM), or (antmaze-u-d, IQL, CS-MLP). There are more missing stars in the table.
- Blue annotations in Table 2 are also significantly wrong, if I understand them correctly. Blue should mark the methods where crowdsourced labels (CS) show better results than the synthetic labels (ST) for the corresponding method. Then, for example, (walker2d-m-r, IQL, CS-MLP) should not be colored in blue, since (walker2d-m-l, IQL, ST-MLP) shows a better score (109.8 > 109.4). I counted 10 instances of such mislabeling in the IQL group alone. This is especially important since the paper claims that "the performance of models trained using crowd-sourced labels (CS) tends to slightly outperform those trained with synthetic labels (ST) in most environments, with significant improvements observed in some cases." Once the blue labels are revised, it is questionable whether this claim still holds. Other offline RL approaches (CQL and TD3BC) show that CS labels are better more consistently, but these approaches perform worse than IQL, so the results for them are correspondingly less important.
- Some CS-TFM results are missing from Table 2. I did not find the explanation for this, maybe I missed it? It would be helpful to see these results in the final version, to better assess the claim of the paper that "CS-TFM exhibits superior stability and average performance compared to CS-MLP".
- For the SMARTS experiment in Section 4.1.2, the paper claims that "the best experimental performance can be obtained simply by crowdsourcing annotations, compared to carefully designed reward functions or scripted teachers." I would say that from the three tasks used one cannot conclude that carefully designed rewards perform worse than crowdsourced annotations. In table 3, the former outperforms the latter on two out of three methods. The claim seems to be made based on average success rate, which only differs by 0.01, less than one standard deviation.
- In Figure 4, speed and torso height are plotted against time. Every 200 steps, the speed attribute value changes (1 to 0.1 and back), so we see the respective changes in speed on the first plot. The relative strength of the "torso height" attribute, however, does not change (stays at 0.1), and the torso height parameters do not change much between the changes. It would be more interesting to see the results where the strength of torso height also changes, so that we can see that it influences the plot.
- Generally speaking, RLHF is most useful in domains where only the humans understand the true reward. This is the case, for example, with "helpfulness" of an LLM, or with "humanness" of the gait of a walking robot. An important evaluation, then, is to see whether the human evaluators prefer the policies trained with an RLHF approach in terms of these hard-to-define measures. This paper, however, does not present such an evaluation. "Humanness" comparisons are collected as shown in Section 4.2, but it is never compared to a policy that is trained without taking humanness into account.

Overall, these weaknesses do not seem to me to be strong enough to motivate a rejection. The dataset and tool contributions are strong, and the problems with baseline evaluations can be clarified in the final version of the paper.

**Questions:**

- In table 2, the results are normalized against an "expert", as the formula in the beginning of Section 4.1.1 shows. How is the expert trained? It is interesting that some of the methods in the table outperform the expert.
- I found the attribute training model in Appendix E.2 confusing. The learned log-preferences $\hat{\zeta}^\alpha$ in eq. (7) probably also need the subscript $\theta$.  The relative strengths of attributes $v^\alpha_{opt}$ are provided as hyperparameters. These strengths are in $[0,1]$. Why, then, in eq. (7) the attributes are checked for being close to $\hat{\zeta}^\alpha$, which is supposed to become the probability of preference according to the respective attribute only after the softmax operation (5)?

---

> ### Author Response · Authors · 2023-11-22
> **Response to Reviewer 3bfZ (Part 1/3)**
>
> We thank the reviewer for the insightful and useful feedback, please see the following for our response.
>
> **[Q1: The claim for a "standardized feedback encoding format along with corresponding training methodologies" might be overly ambitious.]**
>
> Thank you for your valuable suggestion! After careful consideration, we focus on proposing a set of feedback encoding format which are easy to use. We then train them with possible training methodologies based on adapting and improving existing literature, like comparative and attribute feedback in the previous paper version. So we change the statement to **standardized feedback encoding format along with possible training methodologies** in the revision.  (**See Section 3.2, page 5**)
>
> Based on this core idea, we complement the revision with **new crowdsourced labels of keypoint feedback and corresponding experiments** (**See Appendix D.5, page 20**)**,** so the dataset and benchmark contains comparative feedback, attribute feedback and keypoint feedback now. And we provide **more possible training methodologies** **for the all 5/5 feedback types**. (**See Appendix G, page 5**) We will continue to research and expand the platform and datasets for diverse human feedback.
>
> **[Q2: Comparative feedback interface also does not allow the user to indicate the strength of preference]**
>
> We agree with the reviewer that it might be very meaningful to further allow the user to indicate the strength of preference. We have added this option to the platform design through a slider like Anthropic's dataset[1]. We've also updated the description and cited the paper in the **Appendix G.1, page 23:**
>
> > It is worth noting that $y_\text{comp}$ can simply be extended to a finer-grained division of labels including the degree to which A is better than B. For example,  $y_\text{comp} = (0.75, 0.25)$ represents that A is better than B with a smaller degree than (1, 0). Also, the expansion method can simply be trained along the Eq. (4) without any changes.
> >
>
> **[Q3: the paper provides too few details on the way Atari experiments were performed]**
>
> > Related to the previous point, the paper provides too few details on the way Atari experiments were performed. Appendix G and Section 4.1.1 imply that comparative feedback was collected, but Figure 5 (d) in the appendix -- that visual feedback was used instead.
> >
>
> Sorry for the confusion. We begin by clarifying that our atari experiments use comparative feedback for all, which is intended as an implementation of the CNN reward structure. In Figure 5 (d) in the appendix, we demonstrate that we can use the atari environment offline dataset for visual feedback annotation in the Uni-RLHF platform, but we do not provide the annotated dataset for visual feedback and the corresponding experiments due to limited resource in the current version, and we will continue to explore the performance gain of visual feedback in our future work.
>
> > The highlighting in Table 8 should be explained. My guess is that the best of the ST and CS labels is highlighted.
> >
>
> Yes, you are absolutely correct in your understanding! We use highlighting to mark the better one of the ST and CS labels. And, we increase the importance of atari experiments and provide clearer experimental details In the revised main text. (**See Section 4.1.2 Atari Experiments, page 8**)
>
> **[Q4: About some confusion in full results of Table 2]**
>
> We sincerely thank the reviewer for efforts to improve the quality of our papers, In revision, we have carefully examined the data and marks throughout the table, redesigned some of the annotation (blue highlight and *) and re-analysed experimental findings. (**Table 2, page 8**)
>
> > Star annotations in Table 2 are incomplete.
> >
>
> Sorry for the confusion. In the updated revision, we use * to mark the method that works best of all (usually Oracle). Since multiple algorithms are close in effect in many datasets, we take the best performing as the baseline, and if standard deviations in the rest of the algorithms intersect, they are similarly labelled as *.
>
> > Blue annotations in Table 2 are also significantly wrong.
> >
>
> We apologise for the unclear description. In our definition, blue is used to highlighted which of the CS-MLP and CS-TFM methods is better and we've carefully checked in the revised version.
>
> > Some CS-TFM results are missing from Table 2.
> >
>
> We filled in all the missing results in Table 2 and reanalysed them based on complete results.
>
> > About performance comparison using ST and CS.
> >
>
> Thanks for the correction. We modify the description in the revision to argue that CS can achieve competitive performance with ST in IQL backbone, even script teacher (ST) can be regarded as an expert with some degree. And other offline RL approaches (CQL and TD3BC) show that CS labels are better more consistently. We obtain a similar conclusion in atari domain (image observation and discrete action space).

---

> > ### Author Response · Authors · 2023-11-22
> > **Response to Reviewer 3bfZ (Part 2/3)**
> >
> > [**Q5: For the SMARTS experiment in Section 4.1.2, the paper claims that "the best experimental performance can be obtained simply by crowdsourcing annotations, compared to carefully designed reward functions or scripted teachers." I would say that from the three tasks used one cannot conclude that carefully designed rewards perform worse than crowdsourced annotations. In table 3, the former outperforms the latter on two out of three methods. The claim seems to be made based on average success rate, which only differs by 0.01, less than one standard deviation**.]
> >
> > We apologise for the overly ambitious summary and provide more experimental phenomena for in-depth analysis. We provide the design process for the reward items in **Table.5, page 16**, and more evaluation and visualisation of multi indicators in **Appendix D.1**. We find that simple human feedback annotations still work well on the evaluation of more metrics, which is consistent with human intentions. We modified the summary in **Section 4.1.3** to “We empirically demonstrate that we can achieve competitive performance simply by crowdsourced annotations, compared to carefully designed reward functions or scripted teachers.”
> >
> > **[Q6: In Figure 4, speed and torso height are plotted against time. Every 200 steps, the speed attribute value changes (1 to 0.1 and back), so we see the respective changes in speed on the first plot. The relative strength of the "torso height" attribute, however, does not change (stays at 0.1), and the torso height parameters do not change much between the changes.]**
> >
> > Thanks for pointing this out, as you understand, we hope to show with this experiment that by training jointly on multiple attributes strength model, we can allow agents to adjust attributes during motion. Changes in speed follow changes in target attributes very well and are a very good example. And we notice that the instruction-following changes in torso height are not significant enough. In fig.4, 400 steps (0.6→1) and 600 steps (1→0.1) showed changes in line with expectations, while the other changes showed less obvious changes. We've uploaded the full video on our homepage, where the walker switches postures multiple times as the person commands it, which is very interesting, and the task is difficult to design for manual rewards.
> >
> > In the rebuttal phase, We checked and repeated experiments carefully. we found that this may be due to the limited generalisation and alignment capabilities of the context policy of TD3BC baseline, and we expect researchers to extend the baseline (e.g. more stable attribute aligned model or stronger backbone) for an in-depth study based on the attribute feedback.
> >
> > **[Q7: An important evaluation, then, is to see whether the human evaluators prefer the policies trained with an RLHF approach in terms of these hard-to-define measures.]**
> >
> > Thanks for your constructive suggestion. We strongly recognise that the RLHF should primarily address the evaluation of factors that are difficult to quantify, we also raise it in our future work in the **appendix J, page 22**.
> >
> > In the meantime, we add the following experiments:
> >
> > - **Human evaluation:** we conducted human evaluation experiments for the agents trained by attribute feedback. We first collected 3 trajectories with different humanness, which were set to [0.1, 0.6, 1] and invited five human evaluators. The human evaluators performed a blind selection to judge which video is most human-like and which video is least human-like. If the result is consistent with the preset labels, it means that the agents can follow the humanness indicator well. Finally, all people correctly chose the highest humanness trajectory, and only one person chose the lowest humanness trajectory incorrectly. The experimental results confirm that the agents through RLHF training are able to follow the abstraction metrics well. (**See appendix D.4, page 22 and full videos on the homepage**)
> > - **Complex behavioural learning:** we add an extra online RLHF experiment in the revision (**See Appendix E and Fig.9, page 23**). **After totally 200 queries of human annotation, walker can master the continuous multiple front flip fluently**. Through our experiments we demonstrated that Uni-RLHF can access online training to allow agent to learn novel behaviours that are difficult to design reward.

---

> > > ### Author Response · Authors · 2023-11-22
> > > **Response to Reviewer 3bfZ (Part 3/3)**
> > >
> > > **[Q8: In table 2, the results are normalized against an "expert", as the formula in the beginning of Section 4.1.1 shows. How is the expert trained? It is interesting that some of the methods in the table outperform the expert.]**
> > >
> > > For evaluation, we measure expert-normalized scores respecting underlying task reward from the original benchmark (D4RL)[2], which is a fair evaluation widely used in offline reinforcement learning. The calculation is described in detail in the original D4RL benchmark:
> > >
> > > > we normalize scores for each environment roughly to the range between 0 and 100, …, A score of 100 corresponds to the average returns of a domain-specific expert. For AntMaze, we used an estimate of the maximum score possible. For Adroit, this corresponds to a policy trained with behavioral cloning on human-demonstrations and fine-tuned with RL. For Gym-MuJoCo, this corresponds to a soft-actor critic agent (1M samples from a policy trained to completion with SAC).
> > > >
> > >
> > > So you're absolutely right! Some offline reinforcement learning methods can produce policies that outperforms the original data (>100), due to the ability of good generalisability and the emergence of trajectory stitching. Our experimental results also confirm that a similar effect can be obtained by the reward function trained by human feedback.
> > >
> > > **[Q9: Attribute training model in Appendix E.2 is confusing.]**
> > >
> > > Sorry for the confusion. We have made it clearer in the revised version (**See Appendix G.2, page 24, revised version paper**). First, we change $\hat\zeta^{\alpha}$ to $\hat\zeta_\theta^{\alpha}$ . Then, we would like to clarify that relative strengths of desired attributes $v^\alpha_{\text{opt}}$ can be considered a hyperparameter, which constrained between 0 and 1. Next, we normalise all the latent space vectors  $\hat\zeta_\theta^{\alpha}$  in the dataset as $(\hat{\zeta} - \zeta_\text{min})/(\zeta_\text{max} - \zeta_\text{min})$ and calculate the euclidean distance.
> > >
> > > ---
> > >
> > > [1] Bai Y, et al. Training a helpful and harmless assistant with reinforcement learning from human feedback. arXiv:2204.05862, 2022.
> > >
> > > [2] Fu J, et al. D4rl: Datasets for deep data-driven reinforcement learning[J]. arXiv preprint arXiv:2004.07219, 2020.

---

> ### Comment · Reviewer_3bfZ · 2023-11-23
> **The revised version of the paper, other reviewers' comments**
>
> First, I would like to thank the authors for an extensive revision that carefully takes into account the reviewers' suggestions.
> - [regarding the inconsistencies in Table 2 and its analysis in the main text] The new version looks consistent and complete. I agree with the updated analysis in the paper as well.
> - [regarding Atari experiments] I thank the authors for the clarifications in the reply and in the paper.
> - [regarding the more complete discussion of RLHF training methods and the extra keypoint dataset and evaluation] I think this is a valuable addition to the system.
>
> Regarding other reviewer's comments, I believe that the authors' response addresses the concerns well.
> - Reviewer FdzJ points out some weaknesses of the system that are inherent in any RLHF approach where real human feedback is used. I agree with Reviewer FdzJ that the nature of work is incremental, but we both seem to think that the contribution is enough for publication.
> - Reviewer BKHb points out missing experiments for a better understanding of the results, which the authors do provide, with the exception of labels for conversational models. I believe that a system with a focus on non-textual agents is still a sufficient contribution, even if it does not contain evaluation for online RLHF. The revised version goes even further and does include experiments on online RLHF as well.
>
> **Based on these considerations, I am happy to update my rating for the paper from 6 to 8.**

---

### Meta-Review · Area_Chair_v5fn · 2023-12-11

**Metareview:**

### Summary

The paper introduces Uni-RLHF, a comprehensive framework for Reinforcement Learning with Human Feedback (RLHF). It offers a multi-feedback annotation platform, crowdsourced datasets, and baseline implementations for offline RLHF. The proposed platform accommodates diverse feedback types, enabling extensive human feedback data collection across various tasks. Evaluation using this data showcases competitive performance compared to manually designed rewards. Uni-RLHF aims to facilitate the development of robust RLHF solutions by providing open-source platforms, datasets, and baselines. The reviewers highlight its contribution to enabling data collection, sharing datasets, and evaluating RLHF strategies, emphasizing its potential to achieve better RL alignment using human feedback.

### Decision

This paper is well-written and significantly contributes to the RLHF community interested in offline approaches with multiple annotations. The paper is easy to read. The authors have done a tremendous job of addressing the concerns raised by the reviewers. The paper in the last revision, with the added results, addresses most of the concerns raised by the reviewers. Thus, I believe this paper would be of interest for the ICLR community.

**Justification For Why Not Higher Score:**

The paper is interesting and worth accepting. The initial submission of this paper was rushed and caused some confusion among the reviewers. During the rebuttal phase, the authors revised the paper significantly. For this paper to be a spotlight authors would need to include more experiments on more environments and would need to polish the writing.

**Justification For Why Not Lower Score:**

The presented idea is interesting, and the contributions of this paper are significant. The paper is worth publishing, and the reviewer's feedback was mostly positive about this paper.

---

### Decision · Program_Chairs · 2024-01-16

Accept (poster)